# Neural circuits underlying auditory contrast gain control and their perceptual implications

Michael Lohse [1]*, Victoria M. Bajo [1], Andrew J. King [1,2]* & Ben D.B. Willmore [1,2]

Neural adaptation enables sensory information to be represented optimally in the brain despite large fluctuations over time in the statistics of the environment. Auditory contrast gain control represents an important example, which is thought to arise primarily from cortical processing. Here we show that neurons in the auditory thalamus and midbrain of mice show robust contrast gain control, and that this is implemented independently of cortical activity. Although neurons at each level exhibit contrast gain control to similar degrees, adaptation time constants become longer at later stages of the processing hierarchy, resulting in progressively more stable representations. We also show that auditory discrimination thresholds in human listeners compensate for changes in contrast, and that the strength of this perceptual adaptation can be predicted from physiological measurements. Contrast adaptation is therefore a robust property of both the subcortical and cortical auditory system and accounts for the short-term adaptability of perceptual judgments.

[1] Department of Physiology, Anatomy, and Genetics, University of Oxford, Oxford OX1 3PT, UK. [2] These authors contributed equally: Andrew J. King, Ben D. B. Willmore *email: michael.lohse@dpag.ox.ac.uk; andrew.king@dpag.ox.ac.uk

Adaptation to stimulus statistics is a fundamental principle of sensory processing[1–3], which enables the brain to represent sensory information in ways that are computationally efficient[3,4] and robust to noise[5,6]. Certain forms of adaptation to stimulus statistics have been well studied and are known to be present at early sensory processing levels. In the visual system, for example, retinal responses adapt to mean light intensity[7], while in the auditory system, adaptation to mean sound level has been demonstrated at the level of the auditory nerve[8]. Nevertheless, it remains poorly understood how adaptation to higher stimulus statistics changes as a result of hierarchical processing within the sensory systems or how this links to perception.

In both the visual and auditory systems, stimulus contrast—the variability of light or sound level—is a stimulus statistic that results in neuronal adaptation[9–11]. Contrast adaptation may affect multiple neuronal response properties, but is accomplished principally by adjustments in response gain that compensate for the distribution of stimulus levels in a given sensory environment. This specific form of contrast adaptation is known as contrast gain control (or contrast normalization). Visual contrast gain control is implemented at several stages of the visual system[10–17], and is partially guided by corticofugal projections from primary visual cortex[18]. The perceptual consequences of visual contrast adaptation are controversial[19], although one report suggests that this enhances the ability of observers to detect subsequent contrast changes[20]. In the auditory system, however, the relative contributions of subcortical and cortical structures and their role in contrast gain control have not yet been fully elucidated, and it is not known how contrast gain control affects perception.

Contrast gain control is a prominent feature of neuronal responses in the auditory cortex of mice[21] and ferrets[9], but in ferrets it is less robust in the midbrain[6]. Although this implies a primary role for auditory cortex in contrast gain control, other studies have shown that the responses of subcortical neurons are influenced by sensory context[22–28], as well as motor and cognitive demands[29–32]. This raises the possibility that subcortical circuits may also contribute to adaptation to stimulus contrast. Furthermore, descending influences from the cortex need to be considered: manipulation of auditory corticofugal projections can alter the excitability and tuning properties of neurons in both the thalamus[33–35] and midbrain[34–37], but their involvement in adaptation to stimulus statistics remains largely unexplored[27,38].

In this study, we demonstrate the effects of contrast adaptation on human perception, by showing that acuity in a level discrimination task is rapidly adjusted to partially compensate for changes in sound contrast. We also show physiologically that auditory contrast gain control is present to comparable degrees in the lemniscal auditory midbrain, thalamus, and primary auditory cortex of mice, with progressive increases in temporal stability at each ascending processing level. Surprisingly, cortical silencing has no effect on subcortical contrast gain control, despite significant effects on neuronal excitability, suggesting that the midbrain and thalamus implement adaptation independently of cortex. Finally, we show that the strength of perceptual contrast adaptation in humans can be predicted from the physiological contrast adaptation observed in mouse auditory neurons.

## Results

### Human sound level discrimination is modulated by contrast.
To examine the perceptual consequences of changing the contrast of auditory stimuli, we measured the ability of human participants to discriminate the levels of two broadband noise stimuli presented in different contrast environments. The stimuli were 100 ms snippets of noise, separated by 250 ms, and flanked by

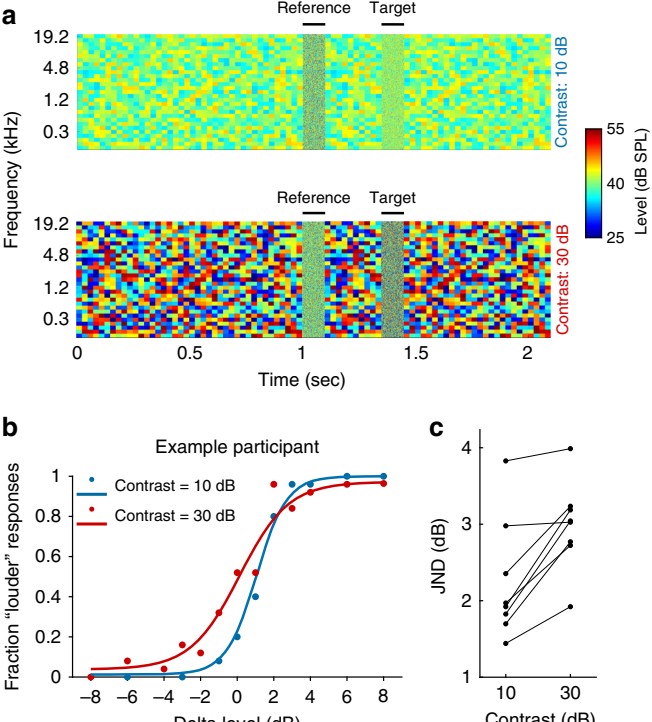

**Fig. 1 Sensitivity to sound level differences in human listeners improves with decreasing auditory contrast. a** Spectrogram illustrating two-alternative forced-choice sound level discrimination task in different contrast environments (dynamic random chords) for human listeners. Participants were instructed to judge whether the target sound (100 ms broadband noise) was "quieter" or "louder" than the reference sound (also 100 ms broadband noise). **b** Examples of psychometric functions from one participant for sound level discrimination in low- (10 dB, blue) and high- (30 dB, red) contrast conditions. **c** Changes in just noticeable difference (JND, difference in dB between 25% and 75% points on psychometric curve) across participants. Source data for **c** are provided as a Source Data file.

dynamic random chords (DRCs) with either 10 or 30 dB contrast (Fig. 1a). We found that level discrimination performance improved when the contrast of the flanking DRCs was low (Fig. 1b), and that this effect was not the result of small contrast-dependent differences in overall sound level that are inherent to the DRC stimuli (Supplementary Fig. 1; see Methods). All participants showed this increase in sensitivity ($t(7) = 5.2$, $p = 0.003$, $n = 8$), as measured by the just noticeable difference (JND, the dB difference between the 25% and 75% points on a fitted psychometric curve; Fig. 1c). The JND increased by a mean of 38.8% between low- and high-contrast conditions (a threefold change in stimulus contrast), corresponding to 28.8% compensation for contrast change.

### Contrast gain control in midbrain, thalamus, and cortex.
In order to understand the role of different sensory processing levels in auditory contrast adaptation, we recorded extracellular activity from neurons in the lemniscal parts of the auditory midbrain (central nucleus of the inferior colliculus, CNIC), thalamus (ventral division of the medial geniculate body, MGBv), and primary auditory cortex (A1) of anesthetized mice while playing complex spectro-temporal stimuli (DRCs, see Methods) with either high- (40 dB) or low- (20 dB) contrast (Figs. 2a–c and 3a, b). We fitted separate spectro-temporal receptive fields (STRFs) to the responses of each neuron to high- and low-contrast stimuli

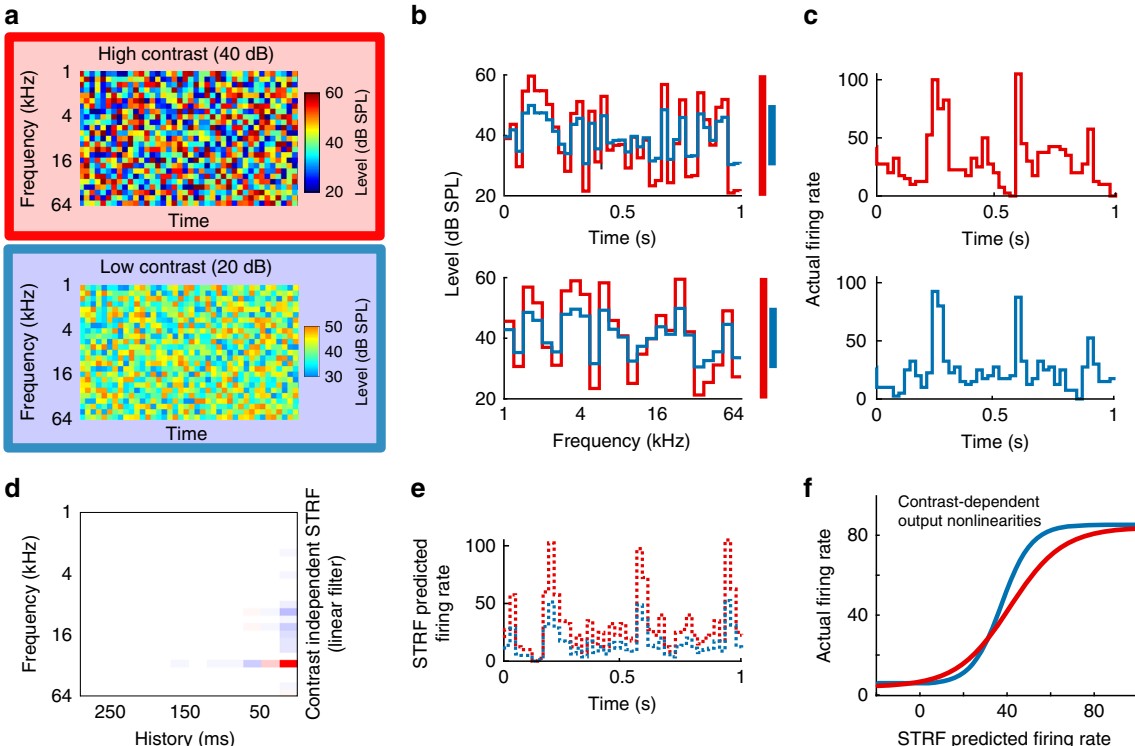

**Fig. 2 Stimulus paradigm for electrophysiological experiments and schematic of linear–nonlinear contrast-dependent model of auditory neurons.**
**a** Spectrograms of snippets (1,000 ms duration) of DRCs with high (red) or low (blue) contrast. **b** Cross-section through an example frequency channel (top) and time point (bottom) of DRCs. Colored bars indicate the sound level range for high (red) and low (blue) contrast. **c** Example peri-stimulus time histograms (PSTHs) during DRC stimulation with high- (top) and low- (bottom) contrast DRCs. **d** Spectro-temporal receptive field (STRF) describing the best-fit linear relationship between stimulus structure and the response of an example neuron (red: positive, white: 0, blue: negative). **e** Example of 1 s of predicted neuronal response to DRCs with high (red) and low contrast (blue), based on the linear STRF model. **f** Sigmoidal contrast-dependent output nonlinearities for an example unit, modeling the relationship between the actual responses of the unit under high- (red) and low- (blue) contrast conditions and the predicted responses of the STRF linear model.

and measured various STRF properties in both conditions. We found that there were significant systematic changes in STRF tuning across the population (Supplementary Fig. 2). However, these changes were small enough that the prediction quality of a single combined STRF was either statistically indistinguishable from (CNIC, MGBv) or better than (A1) the prediction quality of within-condition STRFs (Supplementary Fig. 2). We therefore fitted a single STRF to all the data from each neuron (Figs. 2d–e and 3c) for subsequent analyses. We then fitted an output nonlinearity for each contrast condition (Figs. 2f and 3d). Contrast adaptation in auditory neurons was assessed by comparing the output nonlinearities in high- and low-contrast conditions (see Methods).

As predicted from previous studies[9,21], we found that neurons in A1 exhibited strong contrast gain control—i.e., the slope of the output nonlinearity was adjusted following a change in contrast—and that this gain control largely compensated for the difference in stimulus contrast (Figs. 2f and 3d–e). In auditory cortex, the median degree of compensation was 70.2% ($p = 9.6 \times 10^{-14}$, $n = 106$ units, ten mice, Wilcoxon signed-rank test). Surprisingly, we also found strong compensatory contrast gain control in MGBv (median = 55%, $p = 3.6 \times 10^{-16}$, $n = 136$ units, eight mice) and CNIC (median = 70.8%, $p = 1.7 \times 10^{-64}$, $n = 499$ units, 13 mice; Fig. 3d–e). A Kruskal–Wallis test between contrast gain control in CNIC, MGBv and A1 revealed no significant differences ($p = 0.31$). These results show that neurons in CNIC, MGBv and A1 substantially compensate for changes in stimulus contrast by adjusting the gain of their input–output relationships. These findings did not depend on the assumption that a single

STRF could appropriately describe each neuron (Supplementary Fig. 3). The strength of CGC was not correlated with changes in STRF parameters between contrast conditions, or the magnitude of those changes (Supplementary Fig. 3). These findings were also robust to the specific inclusion criteria used in this study (Supplementary Fig. 4).

Rabinowitz et al.[9] found no difference in contrast gain control in cortical neurons between awake and anesthetized ferrets. We extended this observation by examining whether anesthesia affected contrast gain control in the CNIC. We repeated our recordings in the CNIC of awake, passively listening, head-fixed mice. We found that contrast gain control was robustly present in the CNIC of awake mice (median = 63.9% compensation, $p = 1.2 \times 10^{-50}$, $n = 380$, six mice, Wilcoxon signed-rank test), and indistinguishable in magnitude from that exhibited by CNIC units under anesthesia ($p = 0.1$, Wilcoxon-rank-sum test; Fig. 3d, e). A control experiment confirmed that these effects could not be attributed to small changes in overall sound level between high- and low-contrast stimuli (Supplementary Fig. 5).

We also determined whether the baseline firing rate during DRC stimulation—i.e., the y-offset of the output nonlinearity—was altered by contrast (Fig. 3f and Supplementary Fig. 4). We found that baseline firing rates in CNIC were unaffected by contrast in both anesthetized ($p = 0.46$, Wilcoxon signed-rank test) and awake mice ($p = 0.74$). However, significant decreases in baseline firing rates were measured in both MGBv (−18.5% median change, $p = 9.4 \times 10^{-7}$, Wilcoxon signed-rank test) and A1 (−8.8% median change, $p = 3.1 \times 10^{-9}$) during high-contrast stimulation, potentially providing an additional mechanism to

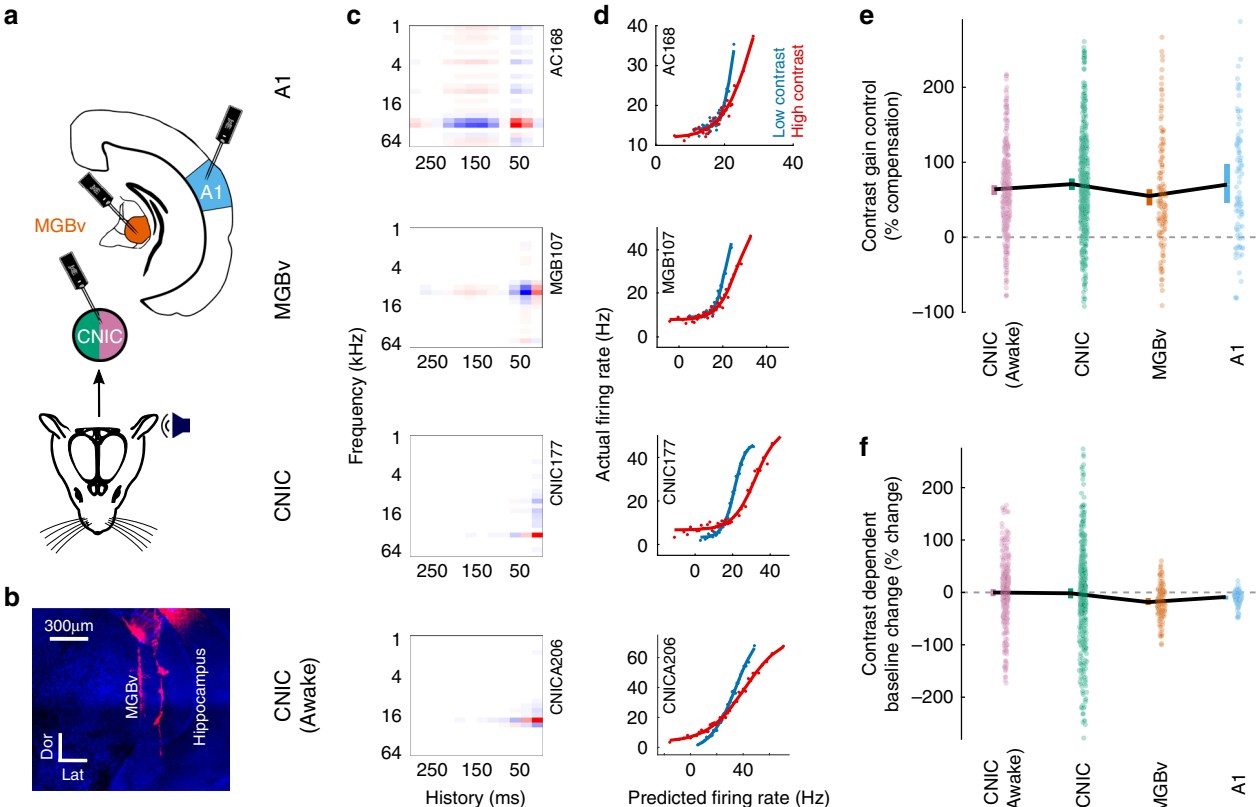

**Fig. 3 Contrast adaptation in the lemniscal auditory pathway. a** Schematic illustrating recordings in A1 and MGBv (under anesthesia) and in the CNIC (in both anesthetized and awake mice). **b** Confocal image showing DiI-coated electrode tracks in the MGBv (Dor, dorsal, Lat, lateral; scale bar, 300 μm). **c** Example STRFs from units recorded in each brain region (red: positive, white: 0, blue: negative). **d** Contrast-dependent output nonlinearities for these same four units. **e** Magnitude of contrast gain control in the auditory pathway, measured as % compensation where 100% would indicate a halving of the gain when the contrast is doubled. **f** Contrast-dependent changes in the baseline activity (y-offset of the output nonlinearity) in the auditory pathway. Colored error bars in **e**, **f**, 95% bootstrapped non-parametric confidence intervals (individual data points (excluding outliers) are displayed next to the corresponding error bars). Source data for **e**, **f** are provided as a Source Data file.

make overall firing rates invariant to contrast at these higher levels of the auditory pathway (Fig. 3f).

**Cortex modulates subcortical excitability and reliability.** Although the auditory cortex has been found to heavily influence the subcortical processing of simple tones[33,35], little is known about its contribution to the representation of complex sounds in the thalamus or midbrain. In order to understand the role of descending corticofugal projections in the implementation of contrast gain control, we first examined the effect of cortical inactivation on the activity of subcortical neurons during continuous DRC stimulation.

Transiently silencing auditory cortex by optogenetic activation of inhibitory neurons (Supplementary Fig. 6) reduced the mean firing rate of MGBv units ($n_{MGBv} = 102$, five mice) during both high-contrast ($-23.6\%$ median change, $p = 4.2 \times 10^{-18}$, Wilcoxon signed-rank test) and low-contrast ($-31.3\%$ median change, $p = 3.4 \times 10^{-18}$) stimulation, as well as the standard deviation of the firing rate across time (high contrast: $-15.8\%$ median change, $p = 7.8 \times 10^{-17}$; low contrast: $-23.1\%$ median change, $p = 2.1 \times 10^{-17}$) (Fig. 4a, b). Similar but weaker effects of cortical silencing were found in the CNIC of awake mice (Supplementary Fig. 7).

Given these strong effects on MGBv activity, and to a lesser degree on CNIC activity, we examined whether corticofugal input influenced the structure of the STRFs in these subcortical regions (Fig. 4c). We measured the effects of cortical silencing on the best

frequency (BF), spectral bandwidth, temporal bandwidth, and on the value of the largest weight in the kernel. We found that silencing auditory cortical activity had no effect on the shape of the STRFs of either MGBv units (Fig. 4d–g) or CNIC units (Supplementary Fig. 7, Supplementary Fig. 8) ($p > 0.05$, Wilcoxon signed-rank tests).

We measured the reliability of neuronal responses by taking the ratio of noise power of the responses to the signal power, NP/SP[39] (see Methods). Surprisingly, the reliability of responses to DRC stimuli was increased (i.e., lower NP/SP) in both MGBv ($-23.8\%$ median change, $p = 1.0 \times 10^{-6}$ Wilcoxon signed-rank test) and CNIC of awake mice ($-11.4\%$ median change, $p = 6.0 \times 10^{-6}$) when cortex was silenced (Fig. 4h and Supplementary Fig. 7). We also found that after silencing auditory cortex, neurons were better described by a linear model in the MGBv (14.9% median change, $p = 8.0 \times 10^{-6}$; Fig. 4i) and in the CNIC of awake mice (4.0% median change, $p = 8.0 \times 10^{-6}$; Supplementary Fig. 7).

These results demonstrate that despite providing a strong excitatory input to MGBv, and to a lesser extent the CNIC, the auditory cortex does not contribute to the receptive field structure of their neurons, but instead influences the reliability and linearity of thalamic responses to complex sounds.

**Subcortical contrast gain control is independent of cortex.** Given the effects of cortical silencing on subcortical responses, it is possible that contrast gain control in MGBv and CNIC neurons

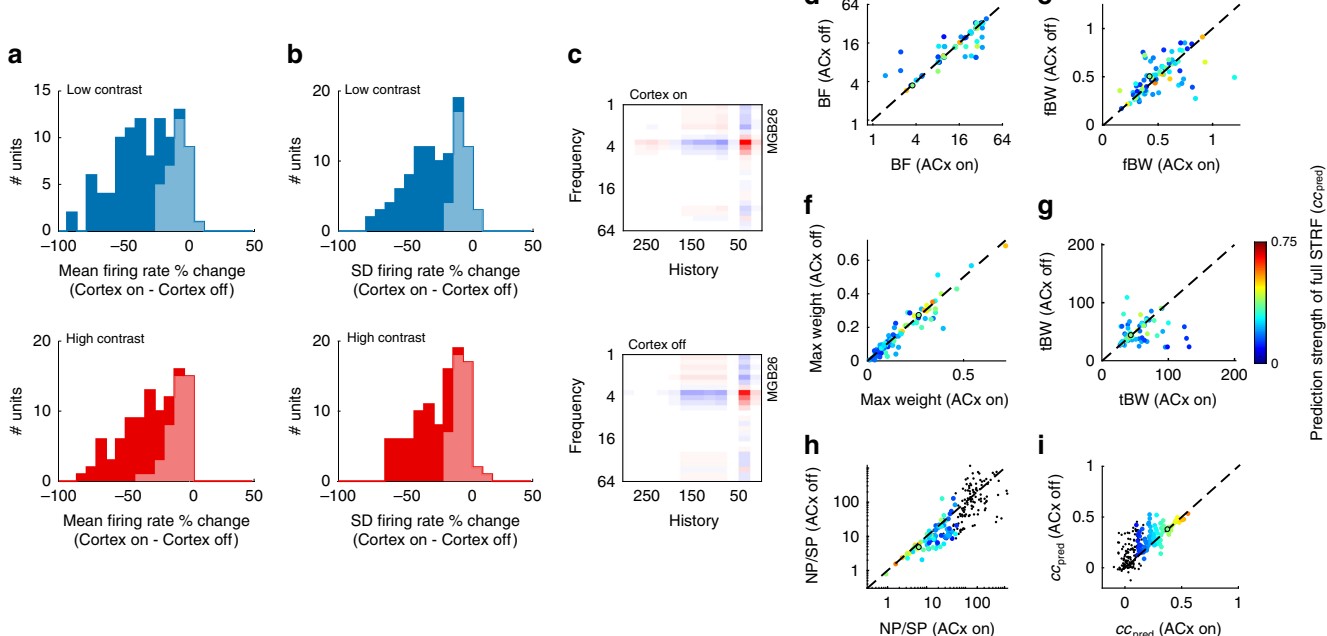

**Fig. 4 Silencing auditory cortex decreases excitability in MGBv, increases reliability and linearity, but leaves STRF parameters unaffected. a** Change in mean firing rate in MGBv during low-contrast (top, blue) and high-contrast (bottom, red) DRC stimulation following optogenetic cortical silencing. **b** Change in standard deviation (SD) of firing rate in MGBv during low-contrast (top) and high-contrast (bottom) DRC stimulation following optogenetic cortical silencing. Light shaded areas in **a** and **b** indicate units that were not significantly modulated by cortical silencing, while dark areas represent units that were affected by cortical silencing ($p < 0.05$, $t$-test). **c** Example STRF (example is marked with black circle in panels **d–i**) of an MGBv unit with auditory cortical activity intact (top), or auditory cortex optogenetically silenced (bottom) (red: positive, white: 0, blue: negative). **d** Comparison of the best frequency (BF), i.e., the largest value of the spectral kernel of the STRF, of MGBv units between recordings made with auditory cortical activity intact (ACx On) or optogenetically silenced (ACx Off). **e** Frequency bandwidth (fBW), i.e., the full-width half-maximum (in octaves) around the BF, of MGBv units with and without cortical silencing. **f** Value of the maximum weight of the STRF of MGBv units with and without cortical silencing. **g** Temporal bandwidth (tBW), i.e., the full-width half-maximum (in ms) around the largest value of the temporal kernel of the STRF, of MGBv units with and without cortical silencing. **h** The ratio between noise and signal power (NP/SP) in the MGBv with and without cortical silencing. **i** Linear model prediction performance within contrast (cross-validated correlation between predicted and actual responses) in the MGBv with and without cortical silencing. Color of points in **d–i** denotes the prediction strength (correlation coefficient) of the model on a cross-validated dataset. Black dots are units excluded from analysis, according to exclusion criteria described in the Methods. Source data are provided as a Source Data file.

might reflect a context-dependent influence of the extensive corticofugal pathways to each of these subcortical structures[40]. Alternatively, subcortical contrast adaptation could be the result of independent computations in the CNIC and/or MGBv. We addressed this directly by optogenetic silencing of auditory cortex while recording from the CNIC and MGBv and presenting DRCs with either high (40 dB) or low (20 dB) contrast (Fig. 5). We fitted separate output nonlinearities to each condition (four conditions) from a linear spectro-temporal prediction across all conditions (cortex silenced or intact, with high- or low-contrast stimuli) (Fig. 5a, b).

We found that subcortical contrast gain control in anesthetized mice was not affected by transient optogenetic cortical silencing. This was the case for units in both MGBv ($p = 0.1$, $n = 99$, five mice, Wilcoxon signed-rank test) and CNIC ($p = 0.5$, $n = 169$, five mice) (Fig. 5b, c). To control for anesthetic state, we carried out optogenetic cortical silencing in awake head-fixed mice while recording from CNIC. Again, we found no effect on contrast gain control in the CNIC ($p_{CNIC\_awake} = 0.3$, $n_{CNIC\_awake} = 129$, three mice) (Fig. 5b, c).

We also examined whether auditory cortex contributes to the effects of contrast on the y-offset in the MGBv. Cortical silencing had a marginal, non-significant effect on this value in MGBv units ($p_{MGBv} = 0.054$, $n_{MGBv} = 99$, five mice), suggesting that the contrast-dependent change in y-offset adaptation may not depend on cortical activity (Fig. 5d). These results therefore

suggest that auditory cortex does not provide the basis for the auditory contrast adaptation (gain control and y-offset adaptation) exhibited by subcortical neurons.

**Contrast adaptation slows along the auditory pathway.** To assess the dynamics of contrast gain control at different levels of the auditory pathway, we collected an additional dataset with recordings (under anesthesia) from CNIC ($n = 155$ units, four mice), MGBv ($n = 56$ units, four mice) and A1 ($n = 73$ units, four mice). We presented DRCs whose contrast switched between high (40 dB) and low (20 dB) values every 2 s. We modeled responses (Fig. 6a) to this switching DRC using an expanded contrast-dependent LN (Linear–Nonlinear) model, where the parameters of the output nonlinearity were allowed to decay exponentially between high- and low-contrast states with a time constant $\tau$.

In the CNIC, time constants were very fast (median $\tau_{CNIC} = 28$ ms), indicating that substantial adaptation occurred during the first chord (duration 25 ms) after each spectro-temporal contrast transition (Fig. 6b). For many CNIC units, the inclusion of an adaptation time constant did not improve predictions over the standard contrast-dependent LN model. This further suggests that adaptation was rapid compared to the chord duration. Adaptation time increased with each ascending sensory processing step (median $\tau_{MGBv} = 79$ ms; median $\tau_{A1} = 175$ ms) (Kruskal–Wallis test, $p < 0.001$), with post-hoc comparisons

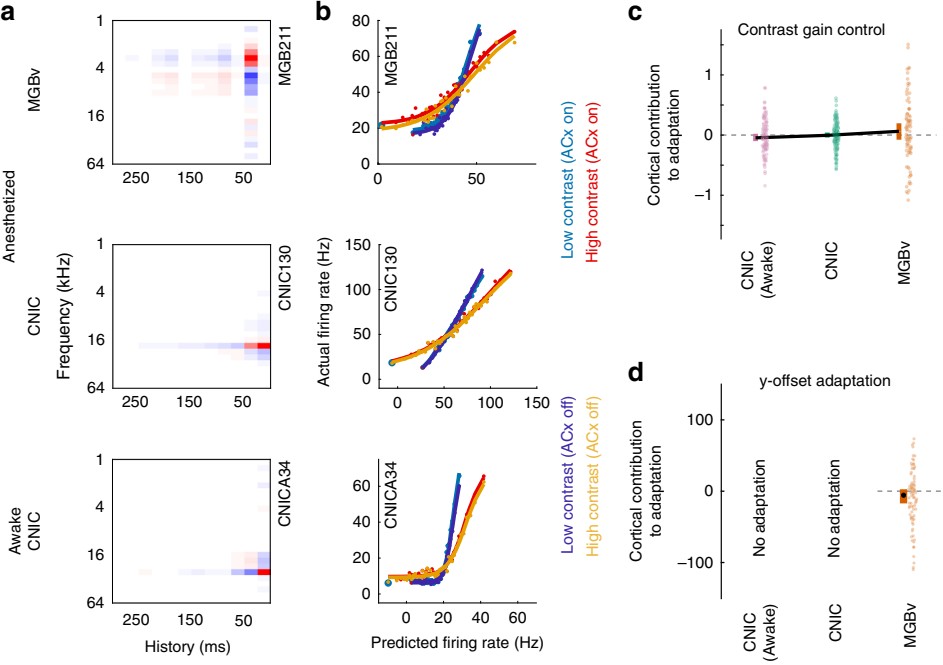

**Fig. 5 Contrast adaptation in the CNIC and MGBv is unaffected by silencing of auditory cortex. a** Examples of spectro-temporal receptive fields of units recorded in MGBv and CNIC of anesthetized mice and in CNIC of awake mice (red: positive, white: 0, blue: negative). **b** The output nonlinearities of the same units during high- and low-contrast stimulation, with or without silencing of cortex. **c** Summary of effects of cortical silencing on contrast gain control in units recorded in MGBv and CNIC of anesthetized mice and CNIC of awake mice; this was quantified as the % gain change with cortex silenced minus the % gain change with cortex intact. **d** Summary of effects of cortical silencing on contrast-dependent y-offset adaptation in the MGBv; this was quantified as % adaptation with cortex silenced minus % adaptation with cortex intact. No contrast-dependent y-offset changes were observed in the CNIC, so the effects of cortical silencing are not shown. **c**, **d** Colored error bars, 95% bootstrapped non-parametric confidence intervals around the medians (individual data points (excluding outliers) are displayed next to the corresponding error bars). Source data for **c**, **d** are provided as a Source Data file.

(Dunn–Sidak corrected) demonstrating significantly longer median adaptation times from CNIC to MGBv ($p < 0.05$) and from MGBv to A1 ($p < 0.05$) (Fig. 6b).

In accordance with this increase in adaptation time from the midbrain to the cortex, the inclusion of an adaptation time constant in the contrast-dependent LN model also became increasingly important. While including adaptation time as a parameter in the contrast-dependent LN model improved the prediction of neural activity in 14.9% of CNIC units, this increased to 25.0% in MGBv, and to more than half the units recorded in A1 (54.8%; black bars in Fig. 6b). A subset of units was estimated to have the maximum time constant allowed by the model (700 ms, because longer time constants could not be reliably estimated using stimuli whose contrast switched every 2 s). This is likely to be a ceiling effect, and suggests that a subset of units have time constants that may be longer than this. Units estimated to have these long time constants were most frequently found in A1.

The progressive increase in time constants might result from differences in the temporal resolution of spectro-temporal representations at different processing levels. Indeed, the temporal bandwidth (estimated as the full-width half-maximum of the temporal kernel in a separable STRF) differed between units recorded at each level (Kruskal–Wallis test, $p = 1.1 \times 10^{-12}$; Fig. 6c). Post-hoc comparisons revealed significantly (Dunn–Sidak corrected) shorter temporal bandwidths in CNIC relative to both A1 ($p < 0.05$) and MGBv ($p < 0.05$). Units in MGBv had intermediate values between CNIC and A1, but these were not significantly different from A1 ($p > 0.05$). However, within each auditory structure, we did not find a correlation between temporal bandwidths and contrast adaptation time constants (Spearman correlation, $p > 0.10$). Thus, although both parameters increase in value along the auditory pathway, temporal bandwidth does not in itself account for the increase in contrast adaptation time.

**Neuronal contrast adaptation accounts for human performance.** Having demonstrated that contrast adaptation can be observed both behaviorally in humans and physiologically in mice, we explored the link between the two. To do this, we developed a model that simulated perceptual judgments in the sound level discrimination task (Fig. 1). This incorporated simulated neural responses, where each simulated neuron was based on the contrast-dependent LN model of a real neuron in CNIC, MGBv, or A1 (Fig. 7a and Supplementary Fig. 9; see Methods).

The strength of perceptual contrast adaptation predicted by the model (mean predicted contrast adaptation: awake CNIC: 19.3%; anesthetized CNIC: 20.2%; MGBv: 17.9%; A1: 21.4%) closely resembled that measured in human participants performing the contrast-dependent sound level discrimination task (28.8%, $n = 8$ participants; Fig. 7b, c). No differences were found between these values (one-way ANOVA, $p = 0.22$), suggesting that the gain control measured at each level of the auditory pathway is sufficient to account for the perceptual adaptation exhibited by human listeners.

## Discussion

Our results demonstrate that auditory contrast adaptation, which has been associated mainly with the auditory cortex[9,21], is exhibited to a similar degree by neurons in lemniscal subcortical structures—the CNIC and MGBv. Moreover, we have shown that this subcortical adaptation is independent of cortical activity. We also found that perceptual thresholds in a sound level

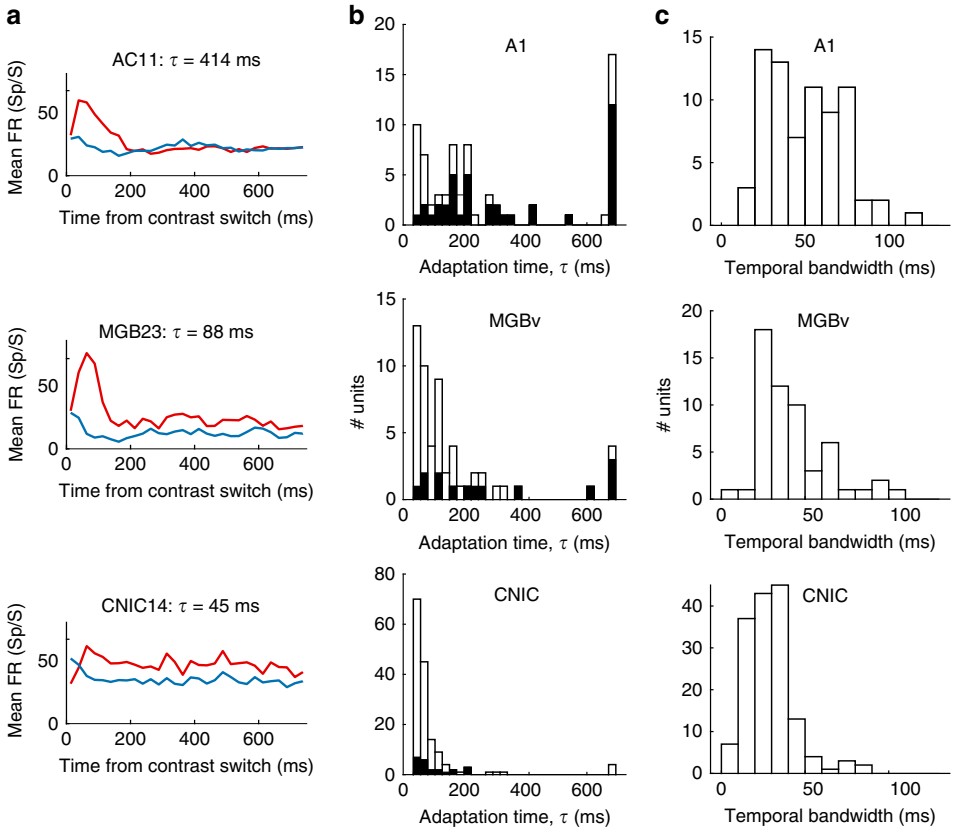

**Fig. 6 Increasing time constants of contrast adaptation along the ascending auditory pathway. a** Mean PSTHs from example units recorded in A1, MGBv, and CNIC after switching from low to high contrast (red) or high to low contrast (blue). The adaptation time constant for these units is given above each plot. **b** Contrast adaptation time constants ($\tau$) for all units recorded using continuously switching contrasts in A1, MGBv, and CNIC. Black bars indicate a subset of these units whose model prediction performance ($cc_{pred}$) was improved by including an adaptation time constant in the contrast-dependent LN model. **c** Temporal bandwidth of these units. Source data for **b**, **c** are provided as a Source Data file.

discrimination task compensate for contrast in a similar way, and that the strength of perceptual contrast adaptation can be predicted from the gain control exhibited by auditory neurons.

Previous work in the ferret has shown that contrast adaptation is weaker and less consistent in the CNIC than in A1[6], and does not consistently compensate for stimulus contrast. In contrast, the results of this study show that compensatory contrast gain control in mice is not purely a cortical computation, but is present to a comparable degree in both the lemniscal auditory midbrain and the thalamus. Although the contrasts used by Rabinowitz et al.[6] were different from those used in the present study, it is possible that this reflects a difference in subcortical computations between mouse and ferret. In both species, however, the data suggest a hierarchy of contrast adaptation, wherein subcortical structures exhibit contrast gain control but in cortex this becomes more consistent across neurons (in ferrets) or more temporally stable (in ferrets and mice). In the visual system, a hierarchy of contrast normalization is present at multiple processing levels from the retina upwards[41]. Similarly, prediction error signals increase along the auditory pathway[42]. Altogether, these findings suggest that some aspects of adaptation to stimulus statistics are organized in a serial fashion in the brain.

It is possible that contrast gain control is also exhibited by neurons in more peripheral structures, particularly as adaptation to mean sound level takes place in the auditory nerve[8]. However, modeling studies suggest that contrast gain control is present to a very limited degree in the auditory nerve[6]. In any case, our results show that auditory subcortical neurons can execute contrast gain control without the involvement of cortical activity. A full

understanding of contrast gain control will therefore require new hypotheses to be developed about the subcortical neural circuitry and mechanisms that underlie this fundamental property of auditory neurons.

Although we found that the overall strength of contrast gain control is similar in CNIC, MGBv and A1, adaptation is not the same at each level of the processing hierarchy. A reduction in baseline firing rate during high-contrast stimulation, which may provide an additional mechanism for making overall firing rates invariant to contrast, is found only in the MGBv and A1. Furthermore, the temporal dynamics of contrast gain control change as we ascend the auditory pathway, suggesting that additional contrast-dependent processing happens at each level. In keeping with Rabinowitz et al.[6], we found that the time constants for auditory contrast gain control become longer at higher levels of the processing hierarchy. This mirrors previous results showing that the temporal integration window for auditory inputs becomes longer from the CNIC, through MGBv, to A1[43,44].

The changes we observe in adaptation time constant cannot be accounted for by temporal bandwidth changes in neuronal STRFs. This suggests that neurons at each processing level may actively adapt to the recent history of stimulus contrast over a range of time scales, rather than merely acting as relays for the transmission of auditory contrast. The progressive increase in the time constant of adaptation along the auditory hierarchy is likely to result in an increasingly stable representation of the auditory environment in the cortex relative to subcortical nuclei. Furthermore, the presence of multiple time scales of adaptation at different levels of the auditory pathway may provide an effective

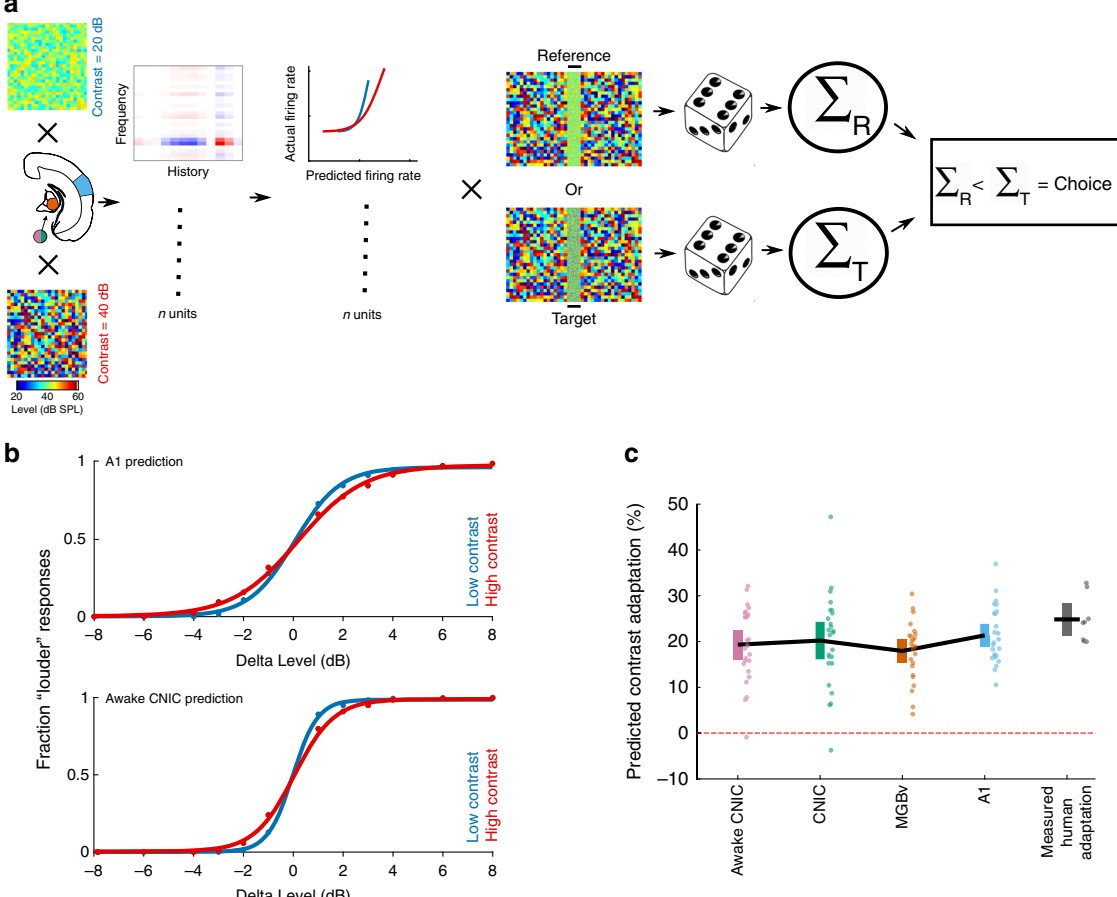

**Fig. 7 The strength of perceptual contrast adaptation can be predicted from contrast adaptation in auditory neurons. a** Schematic of model that uses the neuronal responses to predict performance on a 2-AFC sound level discrimination task (100 ms broadband noise in different contrast environments; see Methods). **b** Psychometric functions produced by the model from A1 units (top) or awake CNIC units (bottom) in low- (20 dB, blue) and high- (40 dB, red) contrast conditions. **c** Predicted strength of contrast adaptation from units recorded in awake CNIC or in CNIC, MGBv, or A1 under anesthesia, compared with measured perceptual contrast adaptation in human listeners. Solid black lines connect mean values after 25 runs of the model (or across the eight participants in the measured human adaptation). Colored error bars denote 95% confidence intervals around the mean (individual data points are displayed next to the corresponding error bars). Source data for **c** are provided as a Source Data file.

means for representing sounds presented in different acoustical environments or tasks. Such diversity of dynamics among different cells also exists for visual contrast adaptation in the retina[45] and adaptation to mean level in the CNIC[26], suggesting that this may be a widespread property of sensory systems.

Contrast gain control in the auditory cortex appears to be a specific case of neuronal normalization wherein the sensitivity of neurons adjusts to compensate for stimulus contrast[41,46]. It has been suggested that normalization is a canonical computation in sensory systems and is present at multiple processing levels[41]. The results presented in this study expand on this idea by demonstrating that contrast gain control is not only a property of neurons in auditory cortex, where it has been studied most extensively[6,9,21,47], but, at least in mice, is equally robust in the CNIC and MGBv. Contrast gain control is therefore established at a relatively early processing level in the auditory pathway.

Our results demonstrate for the first time an important role for the thalamus in contrast adaptation, by both increasing the duration of the adaptation time constants and introducing a subtractive component (y-offset adaptation) that is subsequently inherited by cortex. Neither the contrast gain control nor the subtractive component in contrast adaptation found in the thalamus is dependent on auditory cortical activity. Thus, the

thalamus is an active contributor to contrast adaptation in the ascending auditory pathway, and not merely a relay from the midbrain to the cortex.

The longer adaptation time constants we observe in the cortex suggest that further contrast-related processing happens there. As the representation of sound features changes along the ascending auditory pathway[48], corresponding changes in contrast adaptation may be required at each successive stage. If that is the case, an important question for future research will be whether contrast gain control is implemented via different neural architectures, as has been shown for other neuromodulatory computations[49,50]. Thus, although auditory contrast normalization can be viewed as a canonical computation in the brain, it is unlikely to be implemented by a canonical neural circuit.

Corticofugal projections have previously been shown to have modulatory effects on the excitability and tuning properties of neurons in subcortical nuclei in the auditory[33–37,51,52], visual[53], and somatosensory[54,55] systems. Several studies have reported a net excitatory effect of corticothalamic feedback, which can contribute to changes in receptive field shape[33,35,53,54,56]. In the auditory system, corticofugal modulation has mostly been assessed by measuring spontaneous activity and responses to tones and noise[33–35,56], and evidence for how complex sound processing in the thalamus is affected is sparse. However, recent work

suggests that corticothalamic feedback from layer VI of A1 to MGBv contributes to auditory scene analysis[57]. Furthermore, in the somatosensory system, in vitro recordings have demonstrated that the effects of corticothalamic feedback are dynamic, changing from suppressive to facilitatory depending on stimulation frequency[58], suggesting that corticothalamic input may contribute to context-dependent processing of sensory information.

Our cortical silencing results indicate that while corticofugal inputs have a strong effect on the overall excitability of thalamic neurons (and a weaker effect on the CNIC), the receptive field properties of neurons in these subcortical structures remain unchanged. The reduction in excitability induced by transient optogenetic silencing of the auditory cortex, and the difference in corticofugal effects on CNIC and MGBv, are in accordance with what would be expected from previous studies of the effects of widespread inactivation of A1 on subcortical responses to simple stimuli[59]. However, focal silencing or activation of auditory cortical areas can shift the BF of neurons in both the MGBv and CNIC[34,35]. Manipulating the activity of frequency-specific regions of auditory cortex may therefore have similar effects on the structure of the STRFs acquired from complex sounds, which would be consistent with a potential role for corticofugal feedback in the task-dependent STRF plasticity of auditory midbrain neurons[29].

It has been proposed on the basis of in vitro investigations that corticothalamic feedback provides synaptic noise, which helps thalamic neurons to integrate synaptic inputs more linearly[60,61]. However, by isolating the corticofugal contribution to the representation of ongoing stimuli in vivo, our results suggest that corticofugal activity decreases the linear input–output relationship and the reliability of neuronal responses in the CNIC and MGBv. In the CNIC, this effect of cortical silencing on the transfer function of the neurons appears to depend on wakefulness, but in the MGBv was present even under anesthesia, implying that it is not simply a result of trial-to-trial variability in corticofugal synaptic transmission due to changes in cognitive state.

Although cortical silencing alters the excitability, reliability, and linearity of MGBv and CNIC responses, we found no effect on the strength of contrast gain control. This is consistent with the lack of effect of widespread cortical cooling on adaptation to mean level by IC neurons[27]. However, cortical deactivation does prevent the change in the rate of adaptation by IC neurons following repeated exposure to stimuli with different sound level distributions[27]. It is therefore possible that descending corticofugal inputs might play a role in contrast adaptation in rapidly changing acoustic environments.

The role of corticofugal inputs could be clarified by investigating contrast adaptation in non-lemniscal subdivisions of the auditory thalamus and midbrain. In this study, we aimed to investigate the evolution of contrast adaptation across the primary inputs to A1. However, neurons in non-lemniscal regions of the MGB and IC display stronger stimulus-specific adaptation than their lemniscal counterparts[25,38] and receive stronger inputs from the auditory cortex[38,62]; these neurons may also show correspondingly stronger contrast adaptation, which is under the influence of cortical activity.

The behavioral consequences of adaptation to stimulus statistics in the auditory system have received very little attention. Presenting sounds with interaural level differences[63] or interaural time differences[64] that follow specific statistical distributions results in comparable adaptive changes in the sensitivity of binaural neurons in the brain and in the perceptual sensitivity of human listeners. Furthermore, adaptation to mean level and contrast can improve the decoding of complex sounds from population neuronal activity, potentially providing a mechanism for establishing noise invariance[6].

Our results show for the first time that human auditory perception is subject to a behavioral form of contrast gain control. We also show that the strength of perceptual contrast adaptation in humans is predictable from physiological contrast adaptation in midbrain, thalamic, and cortical auditory neurons in mice. This highlights the importance of adaptation in regulating both neuronal and perceptual sensitivity according to the ongoing statistics of the sensory environment. Furthermore, there is evidence that contrast gain control may mediate the effects of attention on neural processing[65,66]. It would therefore be interesting to determine whether contrast gain control at different levels of the auditory system can be differentially modulated depending on the sensory and behavioral contexts in which sounds occur.

The demonstration in this paper of the widespread and robust nature of auditory contrast adaptation at both physiological and perceptual levels highlights the importance of this adaptive mechanism, and shows that a complex computation with strong implications for behavior can be implemented in subcortical circuitry without the need of cortex.

## Methods

**Mice**. All animal experiments conformed to ethical standards approved by the Committee on Animal Care and Ethical Review at the University of Oxford and were licensed by the UK Home Office (Animal Scientific Procedures Act, 1986, amended in 2012). A total of 39 mice were used in this study. Four strains of male and female mice were used in the electrophysiological experiments: C57BL6/J (Envigo, UK), GAD2-IRES-cre (Jackson Laboratories, USA), VGAT-ChR2-YFP (Jackson Laboratories, USA), and C57BL6/NTac.Cdh23[67]. C57BL6/J, GAD2-IRES-cre, and VGAT-ChR2-YFP were 7–12-weeks old at the time of data collection, and C57BL6/NTac.Cdh23 were 10–20-weeks old at the time of data collection. All experiments were carried out in a sound-attenuated chamber.

**Human subjects**. All procedures conformed to ethical standards approved by the Inter-divisional Research Ethics Committee at the University of Oxford (R52936/RE001). Eight (four male, four female) (plus two additional participants (both male) for the level control experiment) human participants (18–30-years-old) with normal audiometry provided informed consent and participated in the contrast-dependent sound level discrimination study. All experiments were carried out in a sound-attenuated chamber.

**Electrophysiology stimuli**. Stimuli were presented with a Tucker-Davis Technologies (TDT) RX6 Multifunction processor at ~200 kHz. Sounds were amplified by a TDT SA1 stereo amplifier and delivered via a modified Avisoft ultrasonic electrostatic loudspeaker (Vifa) positioned ~1 mm from the ear canal. The sound presentation system was calibrated to a flat (±1 dB) frequency-level response between 500 and 64,000 Hz.

Stimuli consisted of dynamic random chords (DRCs) with individual chords having a duration of 25 ms (including 5 ms on and off ramps) and comprising 25 superposed frequencies, logarithmically spaced between 1000 and 64,000 Hz (1/4 octave intervals). The tones of the DRC were played at sound levels that were randomly drawn from one of two uniform distributions: 30–50 dB sound pressure level (SPL) (low contrast) or 20–60 dB SPL (high contrast). The mean of the distribution was therefore constant, at 40 dB SPL. The logarithmic statistics of the decibel scale have been found to better match the statistics of natural sounds[44,68]. The overall sound level of the DRCs was calibrated to be 79–83 dB SPL. A DRC for any given trial was played for either 40 s or 5 s (5-s trial duration in optogenetic experiments), with inter-trial intervals of 2–10 s. DRCs have previously been used to assess contrast adaptation in the auditory system of ferrets and mice[6,9,21,47].

The overall sound level of high-contrast stimuli was slightly (~3 dB) higher than that of the low-contrast stimuli, due to the nonlinearity inherent in the logarithmic scale. An additional experiment was therefore carried out in which the overall sound levels of DRCs were matched in low- and high-contrast stimuli, at the expense of equality of sound levels of individual tones in the DRCs, to control for possible effects of this small difference in overall sound amplitude (see Supplementary Fig. 5).

**In vivo extracellular recordings**. We carried out extracellular recordings using 32- or 64-channel silicon probes (NeuroNexus Technologies Inc.), in a 4 × 8, 8 × 8, or 2 × 32 electrode configuration. Electrophysiological data were acquired on a Tucker-Davis technologies (TDT) RZ2 BioAmp processor and collected and saved using custom-written Matlab code (https://github.com/beniamino38/benware).

For experiments carried out under anesthesia, mice were anesthetized with an intraperitoneal injection of ketamine (100 mg kg⁻¹) and medetomidine

(0.14 mg kg$^{-1}$). We also administered intraperitoneal injections of atropine (Atrocare, 1 mg kg$^{-1}$) to prevent bradycardia and reduce bronchial secretions, and dexamethasone (Dexadreson, 4 mg kg$^{-1}$) to prevent brain edema. Prior to initial surgery, bupivacain was administered as an analgesic under the scalp. The depth of anesthesia was monitored via the pedal reflex and small additional doses of the ketamine/medetomidine mix were given subcutaneously approximately every 15 min once the recordings started (~1–1.5 h post induction of anesthesia). The dosage of individual top-ups depended on the depth of anesthesia at the time, but corresponded to ~50 mg kg$^{-1}$ h$^{-1}$ of ketamine and ~0.07 mg kg$^{-1}$ h$^{-1}$ of medetomidine. All recordings were performed in the right hemisphere. A silver reference wire was positioned in visual cortex of the contralateral hemisphere, and a grounding wire was attached under the skin on the neck. The head was fixed in position with a metal bar acutely attached with bone cement to the skull over the left hemisphere. We then made 2-mm diameter circular craniotomies above the IC (centered ~5 mm posterior from bregma and ~1 mm lateral from midline), over the visual cortex for auditory thalamic recordings (centered ~3 mm posterior from bregma and ~2.1 mm lateral from midline), and/or over the auditory cortex (centered ~2.5 mm posterior from bregma and ~4.5 mm lateral from midline). Following exposure of the brain, the exposed dura mater was kept moist with saline. The silicon probe was then inserted carefully into the recording site of interest.

In the mouse, the dorsal surface of the IC is not covered by the cortex, and is very distinct[62]. The craniotomies over the IC were always large enough to see the entire exposed IC surface, so we could visually target the probes. We inserted the probe in the center of the IC, and therefore above the CNIC, where the overlying dorsal cortex is relatively thin[62]. We confirmed this by checking for a clear dorsoventral tonotopic gradient in the STRFs that is indicative of this nucleus[69,70]. The tuning widths of the STRFs measured in CNIC are shown in Supplementary Fig. 2. We also estimated frequency response areas using tones, which confirmed the presence of dorso-ventral tonotopic gradients with narrow tuning (data not shown). When we were positioning the electrode array, we observed tightly locked multiunit responses to noise stimuli, characteristic of CNIC neurons, and postmortem inspection of the midbrain confirmed that the probe had indeed been located in the CNIC.

Prior to insertion into auditory thalamus, the probe was coated in DiI (Sigma-Aldrich) for subsequent histological verification of the recording site. Recording sites were confirmed as being located in auditory thalamus if multiunit activity responded to broadband noise and was frequency tuned when the tip of the probe was ~2.5–3.5 mm below the brain surface. Auditory thalamic recordings were subsequently attributed to MGBv by histological investigation of recording sites and by analysis of physiological responses. Based on an immunohistochemical study by Lu et al.[71] on the shape and size of subdivisions of the mouse auditory thalamus, we allocated recording sites to the MGBv if they responded reliably to DRC stimulation on electrode channels < 500 μm from the lateral border of the MGB (see data inclusion criteria).

Finally, A1 was identified by robust neuronal responses to broadband noise bursts, and a caudo-rostral tonotopic axis. Cortical tonotopy was assessed in 4/10 mice by estimating frequency response areas from responses to pure tones on four recording shanks spaced 200 μm apart spanning 600 μm along a rostro-caudal gradient.

For awake recordings in the IC, we chronically implanted a recording chamber under isoflurane (1.5–2% in O$_2$) general anesthesia. The recording chamber consisted of a metal cylinder positioned over a craniotomy, with a lightly attached circular window in order to close the recording chamber. We placed the recording chamber above the IC, together with a head bar and a reference (silver wire) in the contralateral hemisphere. We then fixed the implant to the skull using a dental adhesive resin cement (Super Bond C&B). Following full recovery, on a subsequent day the mouse was head-fixed, the recording chamber was opened, and a sterile recording probe was acutely inserted into the brain via the recording chamber.

**Optogenetic silencing of auditory cortex**. To transiently silence the activity of auditory cortical excitatory neurons, we employed either a transgenic or a viral approach to express ChR2 in auditory cortical inhibitory neurons. *VGAT-ChR2-YFP* mice express ChR2-YFP in GABAergic neurons throughout the adult brain. Optogenetic activation of cortical inhibitory neurons is the most effective available method for inhibiting cortical activity at sub-second time resolution[72] over the time window required for this study, and has been used extensively to transiently silence excitatory activity (including corticofugal outputs) in cortical areas in mice[18,73–75]. Viral injection surgeries were performed under isoflurane (~1.5%) anesthesia, with the animal positioned in a stereotaxic frame (Kopf instruments, USA). For viral transfection, we injected a floxed *AAV5-DIO-ChR2-eYFP* (UNC gene therapy vector core) into auditory cortex of *GAD2-IRES-cre* mice. We injected ~400 nl of virus, spread over three locations (spaced caudal-rostrally ~400 μm apart) at three depths (700, 500, and 300 μm from cortical surface), to ensure widespread expression in auditory cortex (Supplementary Fig. 5). Mice were used for electrophysiological recordings > 4 weeks post injection of virus. This ensured strong expression of ChR2-eYFP in the auditory cortex.

For optogenetic silencing, we exposed the auditory cortex to blue (470 nm) LED light. This was achieved by placement of a 200 μm (*VGAT-ChR2-YFP* experiments) or 1 mm optical fiber (*GAD2-cre* + viral ChR2 experiments) immediately above the

dura mater over the auditory cortex to allow for blue light exposure to ChR2-expressing cells. For silencing of auditory cortical activity during recordings in MGBv or CNIC, we stimulated with blue light at 40 Hz frequency using sinusoidal waves or 15 ms pulses (10 ms gaps). When recording from auditory cortex, we stimulated with blue light at 40 Hz using either sinusoidal waves or 15 ms pulses (10 ms gaps) or constant light. Light power was ~5–7 mW mm$^{-2}$ at the tip of the fiber. We found that light stimulation (40 Hz (sinusoid or pulsed) or constant light) effectively silenced activity in auditory cortical neurons by driving inhibitory neurons for the duration of the DRC stimulation (5 s) (Supplementary Fig. 6).

**Human psychoacoustic experiments**. Stimulus presentation and response collection were performed using PsychoPy 1.85.6[76]. Sounds were presented using a MOTU 828 mkII soundcard and delivered via Sennheiser 650HD headphones in a sound-attenuated chamber. The headphones were calibrated to a flat (±1 dB) frequency-level response between 125 and 19,500 Hz.

Stimuli consisted of broadband noise bursts (100 ms) and dynamic random chords (DRCs) comprising 25-ms duration chords with 29 frequencies logarithmically spaced between 150 and 19,200 Hz. DRCs were constructed with each tone of the DRC being played at levels randomly assigned from a uniform distribution, ranging from 35 to 45 dB SPL (low contrast) or 25 to 55 dB SPL (high contrast) around a fixed mean amplitude of 40 dB SPL. The total sound amplitude of the DRCs was measured to be 64–69 dB SPL. The stimulus for each trial was 1,950 ms long, consisting of 1000 ms of DRC, followed by 100 ms broadband noise (reference: 60 dB SPL), 250 ms of DRC, 100 ms of broadband noise (Target: 52–68 dB SPL), and ending with 500 ms of DRC. The overall sound level of high-contrast stimuli was slightly higher relative to low-contrast stimuli (~4 dB).

A control experiment was also carried out, where the overall sound levels of DRCs were matched in low- and high-contrast stimuli, at the expense of the equality of levels of individual tone levels in the DRCs, to determine whether the small difference in overall sound amplitude between the high- and low-contrast stimuli could account for the JND change with contrast (Supplementary Fig. 1).

**Spike sorting**. We clustered potential neuronal spikes using KiloSort[77] (https://github.com/cortex-lab/KiloSort). Following this automatic clustering step, we manually inspected the clusters in Phy (https://github.com/kwikteam/phy), and removed noise (movement artefacts, optogenetic light artefacts etc.). We assessed clusters according to suggested guidelines published by Stephen Lenzi and Nick Steinmetz (https://phy-contrib.readthedocs.io/en/latest/template-gui/#user-guide).

**Signal power and noise power**. In order to identify units that were continuously responsive to DRC stimulation, we measured the signal power (SP) and noise power (NP) of the neural responses[39]. For all results, unless otherwise specified, we excluded units for which the ratio NP/SP > 60, indicating that these units did not respond reliably to the DRCs on repeated trials.

Where relevant, we also tested how well a linear model described the data, using cross-validation. We fitted spectro-temporal linear filters to 80–90% of the data (training dataset) and tested how well the model predicted the responses on the remaining data (test dataset). Units were excluded if the correlation coefficient (Pearson's $r$) between predicted and real responses in the test dataset was <0.1. These cross-validated prediction values are referred to as $cc_{pred}$, indicating cross-validated correlation between the predicted response and the actual response.

**Spectro-temporal receptive fields**. Neuronal response rates were binned to produce peri-stimulus time histograms (PSTHs) at the same temporal resolution (25 ms) as the chords in the DRCs. To exclude transient onset responses, we excluded the first 500 ms of each stimulus and response. Linear spectro-temporal receptive fields (STRFs, $k_{fh}$) were then estimated to describe the relationship between the PSTHs and the sound levels (in dB SPL) of the tones in the DRCs. The STRFs were constrained to be separable in frequency ($f$) and time history ($h$), i.e. $k_{fh} = k_f \otimes k_h$, and were fitted using maximum likelihood[78]. The separability constraint was used because it reduces the number of parameters that need to be estimated, and can give good STRFs when experimental data are limited[11]. We found that this approach produced acceptable STRFs in all three areas that we recorded from.

For each unit, STRFs were first fitted to data from individual contrast conditions separately, in order to assess contrast-dependent changes in spectro-temporal structure. Subsequently, a single overall STRF was fitted to data from both contrasts, for estimation of contrast-dependent output nonlinearities.

**Contrast-dependent output nonlinearities**. For each contrast condition, we fitted a sigmoid function to the relationship between the actual firing rate of each neuron, $y_t$, and the responses, $z_t$, predicted by the unit's overall STRF[79,80], producing a modelled firing rate $\hat{y}_t$:

$$\hat{y}_t = a + \frac{b}{1 + e^{-(z_t - c)/d}} \qquad (1)$$

by estimating the parameters of the sigmoids in different contrast conditions

($a$ = y-offset, $b$ = y-range, $c$ = x-offset, $b/(4d)$ = gain), we were then able to estimate contrast-dependent changes in the response properties of each unit.

Contrast gain control was measured as percentage compensation in response to a doubling of contrast, where complete (100%) compensation is defined as a halving of gain, and no compensation is defined as no change in gain:

$$\% \text{ compensation} = \frac{C_{\text{low}}(G_{\text{low}} - G_{\text{high}})}{G_{\text{high}}(C_{\text{high}} - C_{\text{low}})} \times 100 \quad (2)$$

For the other variables, we report the percentage change between the values in high ($V_{\text{high}}$) and low conditions ($V_{\text{low}}$), relative to the low-contrast value:

$$\% \text{ change} = \left(\frac{V_{\text{high}} - V_{\text{low}}}{V_{\text{low}}}\right) \times 100 \quad (3)$$

**Contrast-dependent LN model with adaptation time constants.** In order to estimate adaptation dynamics during changing contrast, we used a variation of the contrast-dependent LN model. In this model, the STRF was fixed across conditions, but the parameters of the sigmoid output nonlinearity were fitted separately to high- and low-contrast data, as described above. To estimate the time course of adaptation of the output nonlinearity, we allowed the sigmoid parameters to vary smoothly between their low- and high-contrast values, depending on the exponentially weighted history of recent stimulus contrast. For example:

$$a = a_{\text{low}} + (a_{\text{high}} - a_{\text{low}})\sum \frac{C_t}{n_t}\exp(-t/\tau') \quad (4)$$

where $a_{\text{low}}$ and $a_{\text{high}}$ are the values of $a$ in the low- and high-contrast conditions, respectively, $C_t$ is 0 for low contrast and 1 for high contrast, $t$ indexes the time bins, $n_t$ is the number of time bins, and $\tau'$ is the time constant of the exponential in bins, corresponding to a time constant $\tau$ in ms. The dataset used to estimate the adaptation time course switched between high (40 dB) and low (20 dB) contrast every 2 s. Contrast-dependent LN parameters were estimated from the last second of each contrast presentation. We allowed a maximum $\tau$ of 700 ms, which is the longest value that could be reliably estimated from 2-s epochs. All parameters of the LN model were contrast dependent, and the full model containing LN model parameters from both contrasts along with the estimation of $\tau$ were optimized by gradient decent to minimize the square error between predicted firing rate and the actual firing rate.

In addition to the inclusion criteria used in the LN models for contrast adaptation estimation (see below), we further restricted analysis of time constants to units whose activity was better described by a contrast-dependent LN model than a single contrast-independent model. Consequently, we estimated contrast adaptation time constants only from units that underwent contrast adaptation.

**Psychophysics.** We fitted psychometric functions (https://github.com/wichmann-lab/psignifit) to the probability of participants indicating that the target sound was louder than the reference sound. The JND was estimated as the dB difference between the 25% and 75% points on the psychometric curve. As each listener's sensitivity is inversely proportional to their JND, we assume that the effective gain of the level discrimination process is also inversely proportional to JND, and therefore % compensation can be calculated similarly to the % compensation of contrast gain control above.

**Neurometric behavioral prediction model.** We predicted perceptual contrast adaptation using contrast-dependent LN model simulated responses. We simulated responses to novel broadband noise stimuli of different sound levels (reference: 70 dB SPL, target: 62–78 dB SPL) embedded in low- or high-contrast DRCs (similar to the stimuli used in the psychophysics experiment). This was achieved using response predictions to these novel stimuli from the contrast-dependent LN model estimated from recorded units in the CNIC, MGBv and A1. This was done for every unit included in the analyses of physiological contrast adaptation (separately for each processing level/anesthetic state). For each simulated trial, the simulated response to the broadband noise for each unit was discretized according to a Poisson process, and the simulated onset responses across units were added together. We then asked which noise stimulus elicited most spikes in the simulated trial. If the reference noise elicited fewer spikes than the target noise stimulus, we predicted a "louder" response (Fig. 7a). This process was repeated 500 times for each sound level, in each contrast condition, for estimation of a predicted contrast-dependent psychometric curve from simulated neuronal responses from units in the CNIC (awake or anesthetized), MGBv or A1 (Fig. 7b). We estimated predicted psychometric curves 25 times for each processing level/anesthetic state.

**Reporting summary.** Further information on research design is available in the Nature Research Reporting Summary linked to this article.

## Data availability
The source data underlying Figs. 1c, 3e, f, 4a, b, d–i, 5c, d, 6b, c and 7c are provided as a Source Data File. All relevant data are available on request to, and will be fulfilled by, the lead contact (michael.lohse@dpag.ox.ac.uk).

## Code availability
Matlab code for executing linear–nonlinear models used in this paper can be found on https://github.com/beniamino38/benlib.

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

## Acknowledgements

The research was funded by a Wellcome Trust studentship (WT105241/Z/14/Z) to M.L., and a Wellcome Trust Principal Research Fellowship (WT108369/Z/2015/Z) to A.J.K.

## Author contributions

M.L., B.D.B.W., and A.J.K. conceived and designed the research. M.L. performed and visualized research. M.L. and B.D.B.W. analyzed the data. M.L. and A.J.K. acquired funding for the research. M.L., B.D.B.W., and A.J.K. interpreted the research. M.L., B.D.B.W., V.M.B., and A.J.K. wrote the manuscript.

## Competing interests

The authors declare no competing interests.
