## [Peer Review File · Nature Communications]

Reviewers' comments:

Reviewer #1 (Remarks to the Author):

In the present manuscript, the authors study auditory contrast gain control (sound intensity) at different stages of the auditory pathway (A1, MGBv and CNIC) concluding, via implementation of optogenetic methods to silence cortical regions, that subcortical stages exhibit contrast gain independently from cortical activity. Also, that adaptation time constants become longer at later stages of the hierarchy. And, furthermore, that the strength of perceptual adaptation to contrast in humans can be predicted from modelling the activity from the recorded units.

This is a complete set of studies involving physiological measurements at different levels of the auditory pathway, optogenetics to study causality, and psychophysical measurements in humans, combined in a single, well-written manuscript (albeit with improvable aspects), that could contribute significantly to the field, provided the authors demonstrate that the very first step in their analyses (fitting a single STRF per unit) is fully justified.

Below, find my detailed comments.

Major comments

1) The whole body of results rest in one, non-trivial assumption: that "the differences in tuning were small enough that it was appropriate to fit a single STRF to all the data from each neuron" (lines 90-92). That is, the assessment of contrast adaptation in auditory neurons was performed by comparing the output nonlinearities in high and low contrast conditions, which relate the actual firing rate of a neuron to the predicted responses by the overall STRF. But the claim of being appropriate to fit a single STRF is unsupported. First, no stats are provided. Second, it is unclear which STRF properties and to what degree should be affected in order to decide whether a single STRF should be fitted, or one per condition. Third, a glimpse at Supp.Fig.2 reveals considerable differences across conditions in some STRF properties, at least for several units (the dots in the scatter plot fall far from the diagonal; see the fBW and the max weight, for instance). Therefore, I urge the authors to 1) provide appropriate statistical tests to reveal whether the STRF properties across conditions differ or not (a linear regression fit to the diagonal, which is the expected function the data should follow if the two conditions had no different impact in the activity of the units, seems a reasonable approach, providing adjustment measures and significance values); 2) decide, following a sound rationale, which STRF properties, if any, are allowed to vary significantly across conditions in order to fit a single STRF to the overall activity of a unit (in my opinion, none, as the output non-linearity directly depends on this and it is the measure used to study contrast gain).

2) The authors cite up to 8 papers describing effects of cortical silencing on the shape of STRF of thalamic neurons via corticothalamic projections. But, as they point out, their results show an overall reduction of thalamic activity with no change in STRF properties. However, this claim is again unsupported by the provided data (Wilcoxon signed-rank tests are inappropriate here, where the test should reveal if the data is well adjusted to the diagonal and not if the medians are different), and a quick inspection to Fig.4 reveals, for instance, that the tBW property may have been affected by cortical silencing.

Minor comments

1) The authors could use a vast body of literature about adaptation to stimulus statistics from studies by Malmierca MS. lab to improve their introduction and discussion section. Especially, when dealing with adaptation at different hierarchical levels of the auditory system (Parras, et al., 2017), cortical modulation of adaptation in subcortical regions (Malmierca, et al., 2015), adaptation to frequency vs. intensity in the IC (Duque, et al., 2016), or neurochemical modulation of adaptation (Valdés-Baizabal, et al., 2017; Ayala et al., 2015), among other relevant topics tapped by this group.

2) I would like the authors to discuss further why they recorded from the MGBv and CNIC, when it

has been described that these subdivisions contain neurons that are far less sensitive to stimulus statistics than neurons located at "shell" subdivisions (Ayala, et al., 2015) which, furthermore, receive stronger inputs from auditory cortical areas (Malmierca et al., 2015).

3) Likewise, the authors could further discuss their choice of silencing the AC via inhibitory neurons, while the major corticofugal projections are glutamatergic (Potashner et al., 1988).

4) Line 291: a p value of 0.054 seems insufficient to safely conclude that the contrast dependent change in y-offset adaptation in the MGBv is independent of cortical activity, especially when the overall order of magnitude of the p values across the manuscript is considerably lower. Please, provide CIs and reduce the strength of the conclusions.

5) Line 32: enhances.

6) Unit identity should be noted in each plot depicting activity examples (it would be very illustrative if the example unit (or units) was (were) the same across the different plots and were marked somehow in the STRF properties plots).

7) Line 169: Could it be that the increase in response reliability to DRC stimuli in MGBv and CNIC after silencing represents an "iceberg-effect" (Isaacson & Scanziani, 2011; Ayala, et al., 2016)?

8) Line 443-445: Please, tune down this affirmation, as evidence is indirect (different species, different techniques, different experimental paradigms).

9) Line 672-674: I am confused here. Why did the authors "restrict their analyses of time constants to units whose activity was better described by a contrast-dependent LN model than a single contrast-independent model", when the rest of the study used a single contrast-independent STRF to model the output non-linearities?

Reviewer #2 (Remarks to the Author):

In this study, the authors investigate how contrast gain control is encoded at a neuronal level in several auditory nuclei of the auditory pathway of the mice and correlate such neuronal data with behavioral data acquired with a human perceptual task that requires such gain control for an optimal performance. The authors also try to disentangle the neuronal circuitry related with contrast gain control by inactivating optogenetically auditory cortex while recording in subcortical nuclei of the auditory pathway. The authors continue their main line of research regarding auditory gain control, but going a step further trying to understand the behavioral consequences of such process.

The manuscript is clear and sound, the human behavioral paradigm is simple and smart, and the behavioral results are objectively evaluated. The physiological data recorded is solid and convincing. The study confirms previous findings published by this and other labs and improves the knowledge on the field. Nevertheless, there is a huge gap between the human behavioral experiments and the experiments in mice. The electrophysiological results are really interesting by themselves, but the title claim is a little bit misleading.

Below I detail a major comment and some minor comments that hopefully will improve the final version of the manuscript.

Major comment:

1) As I was saying, considering the current title, I was expecting more evidence for the interaction between the human perceptual intensity discrimination task and the electrophysiological data. If the authors are convinced about the claim of the title, I think they should provide additional experiments where they record neurons in mice while they are performing a 2AFC task -as the one in humans- with two different environments (low and high contrasts). Otherwise, they should tone down such claim and turn the focus to the neuronal and circuitry/optogenetic data as they actually do in the discussion section where 4 out of 5 pages are about electrophysiology.

Minor comments:

1) The authors seem to use "contrast gain control" and "contrast adaptation" as synonyms. In the introduction it seems like "contrast gain control" is the most common phenomenon when considering "contrast adaptation", but no other phenomena are described or referenced in the manuscript. If other effects coexist with "contrast gain control" when talking about "contrast adaptation" they should be described. Otherwise, I suggest picking one of the two concepts for simplification.

2) Page 5, Lines 91-92. Check the references, I think they should refer Supplementary Figure 3.

3) Page 7, Line 131: Introduce reference to Figure 3f.

4) All the data related to the y-offset is really interesting, but such adaptation to the different DRC contrasts (high and low) is unclear. How fast it is? What is the dynamic of such adaptation? Is the baseline firing rate during DRC presentation below the actual spontaneous activity of the neurons (during silence periods)? Some of the answers to these questions may help understanding if such mechanism makes overall firing rates invariant to contrast.

5) Pages 8-9: The concept of reliability (NP/SP) is not introduced when first mentioned in line 169 (Page 8). Moreover, it seems like NP/SP and CCpred are the exclusion criterion the authors use to evaluate the neurons, although they are also a measure by themselves. Thus, it may be better to have subplots h-i before d-g in Figure 4, as I assume the neurons evaluated in subplots d-g are the ones that pass the NP/SP criterion (subplot h). Finally, the explanation in page 8 could also be reorganized accordingly (first NP/SP, then STRF structure).

6) Figure 5: Add anesthetized in the y-axis for CNIC and MGBv plots in "a".

7) Page 12, Line 244-245: Does the model account for different time constants for the two different environmental switches (low-high vs high-low)?

8) Page 12, Line 254: Fig. b should be Fig. 6B.

9) Page 13, Lines 288-289: add "in humans" after behaviorally and "in mice" after physiologically.

10) Page 16, Lines 345-348: As I stated in comment #4, it will be interesting to have the silence spontaneous activity reference.

11) Page 20, Methods section, Electrophysiology: I don't think the number of animals and neurons for the control experiment in Supl. Fig. 4 are stated anywhere in the manuscript. The total amount of animals is not stated in the manuscript neither. Moreover, it is not clear to this reviewer if the same mice were used for the anesthetized recordings in the three different auditory areas.

12) Page 22: Lines 529-531: Considering the positioning of the probe, most of the IC recordings were crossing either rostral and/or dorsal regions of the inferior colliculus. Additional criterion besides the low- to high- frequency gradient may be required to ensure the recordings were in the central nucleus of the IC (frequency width of the STRF, latency...). As the authors are only evaluating the lemniscal pathway, they should ensure the IC recordings are actually in the CNIC.

13) Page 24, Lines 579-581: It looks like the duration of the light pulse (either continuous, pulses or sinusoidal waves) always overlaps the DRC stimulation. Is the optogenetic effect over the spontaneous activity (silence periods) the same as the effect over the DRC stimulation? Is the optogenetic inhibitory effect over the DRC presentation constant? How would the presence of a 100ms broadband noise embedded in a DRC be affected by the optogenetic modulation?

14) Supl. Fig. 1: I think this figure lacks a test comparing discriminability for participants 1 and 2 in corrected vs uncorrected level contrasts as in Supl. Fig. 4.

15) Supl. Fig. 5: It is not stated if there is any difference in optogenetic suppression between the two environments (low and high contrast DRCs).

16) Supl. Fig. 6: I think there is a bad reference to STAR methods.

Responses to Reviewers

The reviewers' comments are in black and our responses are in blue.

Reviewer #1 (Remarks to the Author):

In the present manuscript, the authors study auditory contrast gain control (sound intensity) at different stages of the auditory pathway (A1, MGBv and CNIC) concluding, via implementation of optogenetic methods to silence cortical regions, that subcortical stages exhibit contrast gain independently from cortical activity. Also, that adaptation time constants become longer at later stages of the hierarchy. And, furthermore, that the strength of perceptual adaptation to contrast in humans can be predicted from modelling the activity from the recorded units.

This is a complete set of studies involving physiological measurements at different levels of the auditory pathway, optogenetics to study causality, and psychophysical measurements in humans, combined in a single, well-written manuscript (albeit with improvable aspects), that could contribute significantly to the field, provided the authors demonstrate that the very first step in their analyses (fitting a single STRF per unit) is fully justified.

Below, find my detailed comments.

Major comments

1) The whole body of results rest in one, non-trivial assumption: that “the differences in tuning were small enough that it was appropriate to fit a single STRF to all the data from each neuron” (lines 90-92). That is, the assessment of contrast adaptation in auditory neurons was performed by comparing the output nonlinearities in high and low contrast conditions, which relate the actual firing rate of a neuron to the predicted responses by the overall STRF. But the claim of being appropriate to fit a single STRF is unsupported. First, no stats are provided. Second, it is unclear which STRF properties and to what degree should be affected in order to decide whether a single STRF should be fitted, or one per condition. Third, a glimpse at Supp.Fig.2 reveals considerable differences across conditions in some STRF properties, at least for several units (the dots in the scatter plot fall far from the diagonal; see the fBW and the max weight, for instance). Therefore, I urge the authors to 1) provide appropriate statistical tests to reveal whether the STRF properties across conditions differ or not (a linear regression fit to the diagonal, which is the expected function the data should follow if the two conditions had no different impact in the activity of the units, seems a reasonable approach, providing adjustment measures and significance values); 2) decide, following a sound rationale, which STRF properties, if any, are allowed to vary significantly across conditions in order to fit a single STRF to the overall activity of a unit (in my opinion, none, as the output non-linearity directly depends on this and it is the measure used to study contrast gain).

The reviewer is correct that the analysis presented depends on the assumption that a single STRF can be appropriately fitted to all the data from each unit, and that we did not go far enough in validating this assumption in the manuscript. To address this

thoroughly, we have taken four approaches. These are mentioned in the main text at lines 85-90 and 104-109, and described in detail in Supplementary Figures 2 and 3.

1. One approach for deciding the validity of the single STRF is to compare its predictions with those of the within-condition STRFs, to see whether the single STRF provides a good description of the data. In the previous revision, we made this comparison, and found that the single STRFs (with gain change) predict neural responses just as well as the within-condition STRFs for many units (Supplementary Figure 2, panels E1, E2, E3, E4). We now address this more thoroughly in the expanded description of Supplementary Figure 2. Across the population, the predictions of the single STRFs are either not significantly different from (IC, MGB) or significantly better than (A1) the within-condition STRFs, indicating that the single STRFs are good models of neuronal responses in both conditions, even though small changes in STRF shape do occur. This is stated on lines 85-91.
2. As suggested by the reviewer, we have estimated linear regressions with confidence intervals and performed statistical tests showing how each measure of STRF shape changes between contrast conditions. Bootstrapping demonstrated that, for several comparisons, the regression parameters for the STRF shape changes between high and low contrast follow a bimodal distribution. This indicates that the regression is unstable (even when using robust regression to minimize the effect of outliers) and therefore unsuited as a general test in these comparisons. We have therefore not included this analysis in the manuscript. We have instead included statistics for the more appropriate tests of non-parametric differences and report that there are subtle, but significant, differences in some of the STRF shape parameters (expanded description of Supplementary Figure 2) between contrast conditions. However, as points 1,3 and 4 of the response to this reviewer comment show, these small changes do not systematically bias the ability to predict the firing rate of neurons with a single STRF across the three brain regions and do not affect any of our claims made in the paper.
3. To further validate our conclusion that small changes in STRF parameters do not influence the interpretation of these results, we asked whether changes in individual shape parameters can explain the contrast gain control effects we observe. In the new Supplementary Figure 3, we now plot the strength of contrast gain control against the change in each shape parameter. We find that the strength of contrast gain control is not significantly correlated with changes in any of the STRF shape parameters.
4. Also, we now include a supplementary contrast gain control analysis which does not rest on the assumption that a single STRF is valid. For this analysis, we define gain as the standard deviation of the STRF coefficients. This is a simple measure of how much a unit will respond to sound, regardless of the shape of the STRF. We measured the gain of the within-condition STRFs, G_{low} and G_{high} . We then calculated the gain ratio, G_{low} / G_{high} , which is a measure of the strength of contrast gain control, and used this to calculate the % compensation as before. The results are shown in Supplementary Figure 3b, and these are very similar to the results of the main analysis (Supplementary Figure 3a). This analysis is not as rigorous as the main contrast gain control analysis, but it shows that our core results are not dependent on the assumption that a single STRF is adequate.

In the course of revising this section of the paper, we have improved two other aspects:

1. We have clarified that the change in maximum STRF coefficient is a simple measure of gain, and that we should expect this to change between contrast conditions, unlike the other measures of STRF shape (expanded description of Supplementary Figure 2). This is what we found.
2. We previously measured the maximum coefficient value of the frequency kernel only. However, it would be more appropriate to measure the maximum coefficient value over the entire STRF, so we have now amended these values accordingly where they occur (Figure 4f, Supplementary Figures 2 & 7).

2) The authors cite up to 8 papers describing effects of cortical silencing on the shape of STRF of thalamic neurons via corticothalamic projections. But, as they point out, their results show an overall reduction of thalamic activity with no change in STRF properties. However, this claim is again unsupported by the provided data (Wilcoxon signed-rank tests are inappropriate here, where the test should reveal if the data is well adjusted to the diagonal and not if the medians are different), and a quick inspection to Fig.4 reveals, for instance, that the tBW property may have been affected by cortical silencing.

As suggested by the reviewer, we have performed regression analyses for the data shown in Figure 4d-f. The results of these regressions are shown in Supplementary Figure 8. In all cases, the diagonal is within the confidence intervals. This supports the statement in the paper that subcortical STRF shape is left unaffected by cortical silencing.

Several of the papers we cite used focal silencing, which, like focal stimulation, will modulate the activity of selective regions within the subcortical targets of corticofugal projection neurons. By contrast, our aim was to silence the whole of A1, which would be expected to have a uniform effect on the subcortical targets. We agree that some studies have reported that cortical cooling can alter subcortical response properties (e.g. Nakamoto et al. (2008) reported that cortical cooling affects the sensitivity of IC neurons to binaural spatial cues), but optogenetic silencing is a considerably more refined approach to investigating the role of cortical feedback. Given the expanded analysis of the effects on subcortical STRFs, as well as our data on the excitability and reliability of these neurons, we believe that our conclusions are robust. We have provided more justification for the optogenetic method we used on lines 519-523, and included new text in the Discussion where we speculate about the possible effects of focal silencing or inactivation of A1 on subcortical contrast gain control on lines 348-353.

Minor comments

1) The authors could use a vast body of literature about adaptation to stimulus statistics from studies by Malmierca MS. lab to improve their introduction and discussion section. Especially, when dealing with adaptation at different hierarchical levels of the auditory system (Parras, et al., 2017), cortical modulation of adaptation in subcortical regions (Malmierca, et al., 2015), adaptation to frequency vs. intensity in the IC (Duque, et al., 2016), or neurochemical modulation of adaptation (Valdés-Baizabal, et al., 2017; Ayala et al., 2015), among other relevant topics tapped by this group.

Thank you for suggesting these references, and we apologise for omitting them before. We have added the most relevant references at appropriate places in the text. In particular, we have reworded the introduction where we refer to the effects of sensory context on the response properties of subcortical neurons, and added the most appropriate references from the Malmierca, Linden and Nelken labs (lines 40-47). We have also added the Parras et al paper to the section of the Discussion dealing with hierarchical processing (lines 269-272).

2) I would like the authors to discuss further why they recorded from the MGBv and CNIC, when it has been described that these subdivisions contain neurons that are far less sensitive to stimulus statistics than neurons located at “shell” subdivisions (Ayala, et al., 2015) which, furthermore, receive stronger inputs from auditory cortical areas (Malmierca et al., 2015).

We now address this in the Discussion (line 370-377) -- we wanted to investigate the evolution of contrast gain control in the areas that provide the principal input to A1, i.e. lemniscal regions of the auditory midbrain and thalamus. It would indeed be interesting to look in future work at the “shell” subdivisions of these structures, as well as secondary areas of auditory cortex, and we specifically raise the possibility that the shell of the IC may show stronger contrast adaptation.

The present study, however, focuses on the emergence within the central auditory pathway of contrast gain control, which is a prominent feature of A1, so it is entirely appropriate to have investigated this in MGBv and CNIC. Even if contrast adaptation is widespread in non-lemniscal areas, we would note that our neuronal recordings from lemniscal regions of the mouse auditory pathway are sufficient to account for the human psychophysical data reported here.

3) Likewise, the authors could further discuss their choice of silencing the AC via inhibitory neurons, while the major corticofugal projections are glutamatergic (Potashner et al., 1988).

We wanted to silence excitatory activity in auditory cortex effectively and with sub-second precision, in order to demonstrate whether or not cortical activity affected subcortical contrast gain control and other subcortical processes. Given that our results show that cortical processing does not influence subcortical contrast gain control, it was important to ensure maximum suppression of excitatory cortical activity, as well as to control for trial-to-trial variability. Currently, the most reliable method for fast (sub-second onset and offset) and robust silencing of excitatory cortical activity in mice is optogenetic activation of cortical inhibitory neurons (Guo et al., 2014, Nature; Li et al., 2019, bioRxiv; also see Supplementary Figure 6 for demonstration of strong suppression of excitatory auditory cortical activity in the current paper).

Direct optogenetic inhibition of excitatory cortical cells (i.e. using halorhodopsins or ArchT) does not suppress excitatory activity to the same degree, and would require much stronger light powers (Li et al., 2019), which would potentially cause tissue heating confounds with the multi-second (5 seconds in the current study) inhibition needed to study auditory contrast gain control (Stujenske et al., 2015, Cell Rep). Other approaches for reversible silencing of cortical activity, such as pharmacological (e.g. muscimol) methods, chemogenetic methods (e.g. hM4Di), or cooling techniques, do not provide the temporal resolution required for the current study. We have clarified our choice and cited Li et al (2019) on lines 519-523.

4) Line 219: a p value of 0.054 seems insufficient to safely conclude that the contrast dependent change in y-offset adaptation in the MGBv is independent of cortical activity, especially when the overall order of magnitude of the p values across the manuscript is considerably lower. Please, provide CIs and reduce the strength of the conclusions.

We agree that we stated this conclusion too strongly. We have softened the conclusions and 95% bootstrapped confidence intervals are shown in Figure 5d (lines 185- 87).

5) Line 32: enhances.

Fixed (now line 34)

6) Unit identity should be noted in each plot depicting activity examples (it would be very illustrative if the example unit (or units) was (were) the same across the different plots and were marked somehow in the STRF properties plots).

As requested, we have added unit identity to all examples shown in Figure 3, Figure 4, Figure 5 and Figure 6, and clarified in the legends that the examples of STRF and nonlinearities indeed belong to the same example units. In Figure 4, we also indicate which the example unit is on the scatterplots showing STRF tuning parameters (panels d-i).

7) Line 169: Could it be that the increase in response reliability to DRC stimuli in MGBv and CNIC after silencing represents an “iceberg-effect” (Isaacson & Scanziani, 2011; Ayala, et al., 2016)?

We are reluctant to speculate about this because it is not clear whether an iceberg effect would result in the observed change in response reliability. Isaacson & Scanziani and Wehr & Zador suggest that the timing of inhibitory and excitatory drive affects reliability because action potentials occur relatively reliably in the window where excitation precedes inhibition. In our case, cortical input to MGBv is reduced but we do not know whether its timing is affected, and it is therefore unclear how this would affect reliability in MGBv.

8) Line 443-445: Please, tune down this affirmation, as evidence is indirect (different species, different techniques, different experimental paradigms).

The reviewer is correct to draw attention to this; we have rephrased this more carefully (lines 385-388).

9) Line 672-674: I am confused here. Why did the authors “restrict their analyses of time constants to units whose activity was better described by a contrast-dependent LN model than a single contrast-independent model”, when the rest of the study used a single contrast-independent STRF to model the output non-linearities?

We apologise that this was not very clear. In the time course analysis (as in the other analyses), we used a single contrast-independent STRF for each unit. The only part of the model that changed between contrast conditions was the output nonlinearity, whose four parameters were fitted separately to data from high- and low- contrast conditions. For some units, the output nonlinearities were very similar in both conditions, indicating that little contrast adaptation took place. For these units, it would not be meaningful to estimate the time course of contrast adaptation. We therefore excluded such units from the time course analysis using a criterion that the model which includes contrast adaptation of the output-nonlinearity should predict the neuronal responses better than a fixed model with no contrast adaptation. We have clarified this on at lines 625-629.

Reviewer #2 (Remarks to the Author):

In this study, the authors investigate how contrast gain control is encoded at a neuronal level in several auditory nuclei of the auditory pathway of the mice and correlate such neuronal data with behavioral data acquired with a human perceptual task that requires such gain control for an optimal performance. The authors also try to disentangle the neuronal circuitry related with contrast gain control by inactivating optogenetically auditory cortex while recording in subcortical nuclei of the auditory pathway. The authors continue their main line of research regarding auditory gain control, but going a step further trying to understand the behavioral consequences of such process.

The manuscript is clear and sound, the human behavioral paradigm is simple and smart, and the behavioral results are objectively evaluated. The physiological data recorded is solid and convincing. The study confirms previous findings published by this and other labs and improves the knowledge on the field. Nevertheless, there is a huge gap between the human behavioral experiments and the experiments in mice. The electrophysiological results are really interesting by themselves, but the title claim is a little bit misleading.

Below I detail a major comment and some minor comments that hopefully will improve the final version of the manuscript.

Major comment:

1) As I was saying, considering the current title, I was expecting more evidence for the interaction between the human perceptual intensity discrimination task and the electrophysiological data. If the authors are convinced about the claim of the title, I think they should provide additional experiments where they record neurons in mice while they are performing a 2AFC task -as the one in humans- with two different environments (low and high contrasts). Otherwise, they should tone down such claim and turn the focus to the neuronal and circuitry/optogenetic data as they actually do in the discussion section where 4 out of 5 pages are about electrophysiology.

We agree with the reviewer that the original title is too strong, given that we don't show the results of combined physiology and behaviour in mice. We would like to do this experiment in the future, but we think it is critical to show that a physiological phenomenon, such as contrast gain control, is relevant to human perception. We have therefore retitled the paper as the reviewer suggests. The new title is "Neural circuits underlying auditory contrast gain control and their perceptual implications."

Minor comments:

1) The authors seem to use “contrast gain control” and “contrast adaptation” as synonyms. In the introduction it seems like “contrast gain control” is the most common phenomenon when considering “contrast adaptation”, but no other phenomena are described or referenced in the manuscript. If other effects coexist with “contrast gain control” when talking about “contrast adaptation” they should be described. Otherwise, I suggest picking one of the two concepts for simplification.

We agree that the manuscript did not distinguish clearly enough between these two terms. We do want to maintain this distinction -- where “contrast adaptation” refers to any adaptation which results from changes in contrast, such as changes in baseline firing rate (as demonstrated in Figure 3), and “contrast gain control” refers specifically to contrast-dependent changes in neuronal gain. We have therefore rewritten the relevant part of the Introduction (lines 25-30) to clarify this distinction.

2) Page 5, Lines 91-92. Check the references, I think they should refer Supplementary Figure 3.

We have updated the references to the figures in this paragraph to reflect the more detailed explanation of why a single combined STRF was used, and to correct a reference to Figure 3c to Figure 3d. This paragraph refers to Supplementary Figure 2 in the appropriate places. There is no need to refer to Supplementary Figure 3 here, but it has been added to the new text at the end of the paragraph (lines 104-109).

3) Page 7, Line 131: Introduce reference to Figure 3f.

We have added a reference to Fig. 3f and the new Supplementary Fig. 4d (lines 121-122).

4) All the data related to the y-offset is really interesting, but such adaptation to the different DRC contrasts (high and low) is unclear. How fast it is? What is the dynamic of such adaptation? Is the baseline firing rate during DRC presentation below the actual spontaneous activity of the neurons (during silence periods)? Some of the answers to these questions may help understanding if such mechanism makes overall firing rates invariant to contrast.

We agree that this is interesting. However, the focus of this study is the change in neuronal gain, and the data collected are not well suited to estimating separate time constants for the y-offset changes and the other parameters of the sigmoid. Similarly, we did not collect spontaneous activity in this experiment, so, unfortunately, we cannot directly answer the question of whether baseline activity during DRC stimulation is lower than the spontaneous rate.

However, we have been able to address the question of firing rate invariance. To do this, we measured the range of firing rates (5th and 95th percentiles) of each unit during high and low contrast stimulation. We now show these results in Supplementary Figure 4d. The median of the 5th percentile values for all areas is 0 spikes per second for both high and low contrast conditions, while the median of the 95th percentile values is slightly higher in the high contrast condition than in the low contrast condition. This difference is significant in all recorded areas and both awake and anaesthetised CNIC. Finally, we find that the 95th percentile values for MGBv and A1 units are significantly more invariant to contrast changes than in CNIC (both

awake and anaesthetised). Overall, these results show changes in firing rate that are relatively small (compared with the doubling of stimulus contrast), but are nonetheless consistent with the gain changes we measured using the STRFs. We have added this to the Results (lines 121-122).

5) Pages 8-9: The concept of reliability (NP/SP) is not introduced when first mentioned in line 169 (Page 8). Moreover, it seems like NP/SP and CCpred are the exclusion criterion the authors use to evaluate the neurons, although they are also a measure by themselves. Thus, it may be better to have subplots h-i before d-g in Figure 4, as I assume the neurons evaluated in subplots d-g are the ones that pass the NP/SP criterion (subplot h). Finally, the explanation in page 8 could also be reorganized accordingly (first NP/SP, then STRF structure).

We agree that NP/SP needed more explanation when first used, and so we have added a brief introduction to this in the Methods, and a reference to the Sahani and Linden paper on lines 152-153. We think that the resulting section is now sufficiently clear that the proposed reorganisation of this figure is not now needed.

6) Figure 5: Add anesthetized in the y-axis for CNIC and MGBv plots in “a”.

Fixed.

7) Page 12, Line 244-245: Does the model account for different time constants for the two different environmental switches (low-high vs high-low)?

This experiment was not optimized to estimate separate time courses of the change in gain for the two transitions. We have analysed the data to address this, but the number of valid units in the analysis is too small for a compelling result. For the reviewer’s interest, we find that the transition from high to low contrast tends to be slower than the transition from low to high contrast, as reported for ferret A1 by Rabinowitz et al. (2011), but for the above reason we have not added this to the text.

8) Page 12, Line 254: Fig. b should be Fig. 6B.

Fixed (line 210).

9) Page 13, Lines 288-289: add “in humans” after behaviorally and “in mice” after physiologically.

Fixed (lines 236-237). We have also made a similar change to the end of the Introduction (lines 56-58).

10) Page 16, Lines 345-348: As I stated in comment #4, it will be interesting to have the silence spontaneous activity reference.

As noted above, we unfortunately did not collect spontaneous activity when we were presenting DRCs in this experiment.

11) Page 20, Methods section, Electrophysiology: I don't think the number of animals and neurons for the control experiment in Supl. Fig. 4 are stated anywhere in the manuscript. The total amount of animals is not stated in the manuscript neither. Moreover, it is not clear to this reviewer if the same mice were used for the anesthetized recordings in the three different auditory areas.

This is now Supplementary Figure 5, which shows data from 65 units recorded in 2 mice. These 2 mice were also used for the estimation of the contrast gain control shown in Figure 3. The number of mice and units is now provided in every figure. A total of 39 mice were used in this study. This information has been added to the Methods section (lines 407-408).

12) Page 22: Lines 529-531: Considering the positioning of the probe, most of the IC recordings were crossing either rostral and/or dorsal regions of the inferior colliculus. Additional criterion besides the low- to high- frequency gradient may be required to ensure the recordings were in the central nucleus of the IC (frequency width of the STRF, latency...). As the authors are only evaluating the lemniscal pathway, they should ensure the IC recordings are actually in the CNIC.

In the mouse, the dorsal surface of the IC is visible, and very distinct (see Figure 1 of Barnstedt et al., 2015). The craniotomies over the IC were always large enough to see the entire exposed IC surface, so we could visually target the probes. We always inserted the probe in the center of the IC, and therefore above the CNIC. Furthermore, the overlying dorsal cortex in this region is relatively thin (Barnstedt et al., 2015). We confirmed the CNIC location of the electrode array by checking for a clear dorso-ventral tonotopic gradient in the STRFs. The tuning widths of the STRFs measured in CNIC are shown in Supplementary Figure 2-C1. We also estimated frequency response areas using tones, which confirmed the presence of dorso-ventral tonotopic gradients with narrow tuning (data not shown). When we were positioning the electrode array, we observed tightly locked multiunit responses to noise stimuli, characteristic of CNIC neurons, and post-mortem inspection of the midbrain confirmed that the probe had indeed been located in the CNIC. We now clarify this in the Methods (lines 479-490).

13) Page 24, Lines 579-581: It looks like the duration of the light pulse (either continuous, pulses or sinusoidal waves) always overlaps the DRC stimulation. Is the optogenetic effect over the spontaneous activity (silence periods) the same as the effect over the DRC stimulation? Is the optogenetic inhibitory effect over the DRC presentation constant? How would the presence of a 100ms broadband noise embedded in a DRC be affected by the optogenetic modulation?

For this dataset, we only acquired data during the presentation of the DRCs, which means we do not have data showing the optogenetic effect on spontaneous activity for these recordings.

As part of a different study involving optogenetic silencing of A1, we have measured both tone and noise evoked responses and spontaneous activity in A1 (Figure below panels A-D) and MGB (Figure below panels E-F). We cannot include this figure in the current manuscript because it is part of another article submission for which DRCs

were not presented. However, we show it below for the benefit of the reviewer. This figure shows that optogenetic suppression of auditory cortex silences evoked responses (panel C) and spontaneous activity (panel D) in cortex, as well as decreasing spontaneous activity (panel E) and tone responses in MGB (panel F):

To address the important question of whether the cortical suppression is constant during DRC stimulation, we have calculated the optogenetic suppression in auditory cortex in three different epochs of DRC stimulation (beginning, middle and end of DRC stimulation) and find strong suppression of cortical activity throughout (<5% median activity remaining compared to control activity (i.e., no optogenetic stimulation) in all epochs). We now state this in the legend of Supplementary Fig. 6.

14) Supl. Fig. 1: I think this figure lacks a test comparing discriminability for participants 1 and 2 in corrected vs uncorrected level contrasts as in Supl. Fig. 4.

These two experiments were conducted with different groups of participants, so it is not possible to make this direct comparison. We suspect our original labelling (where we used "Participant 1" etc) in Supp. Fig. 1 gave the erroneous impression that these were the same subjects as in the main experiment. We have relabeled this figure accordingly. We have added a statistical test showing that the JNDs in the control experiment are not significantly different from those in the main experiment ($p = 0.3$, t -test; legend of Supplementary Fig. 1).

15) Supl. Fig. 6: It is not stated if there is any difference in optogenetic suppression between the two environments (low and high contrast DRCs).

There is no difference in optogenetic cortical suppression between high and low contrast conditions. We have added a small panel and legend to Supplementary Figure 6d to show that suppression of cortical units is not significantly different between low and high contrast conditions ($p = 0.13$, signed-rank test).

16) Supl. Fig. 6: I think there is a bad reference to STAR methods.

Fixed.

REVIEWERS' COMMENTS:

Reviewer #1 (Remarks to the Author):

I would like to thank the authors for their thorough answers to my comments. I believe they strongly validated by several means the assumption that a single STRF can appropriately be fitted to all the data from each unit, which was my major concern. The authors satisfactorily addressed the rest of issues I raised and, therefore, I have no further comments and endorse the publication of the manuscript.

Reviewer #2 (Remarks to the Author):

The authors have addressed all my questions and the paper is sound. I don't have any more questions. The manuscript is ready for publication.

Daniel Duque